# Assessing soil salinity dynamics using time-lapse electromagnetic conductivity imaging

Maria Catarina Paz [1,2], Mohammad Farzamian [1,3], Ana Marta Paz [3], Nádia Luísa Castanheira [3], Maria Conceição Gonçalves [3], Fernando Monteiro Santos [1]

5 [1]Instituto Dom Luiz, Faculdade de Ciências da Universidade de Lisboa, Campo Grande, Edifício C1, Piso 1, 1749-016 Lisboa, Portugal

[2]CIQuiBio, Barreiro School of Technology, Polytechnic Institute of Setúbal, Rua Américo da Silva Marinho, 2839-001 Lavradio, Portugal

[3]Instituto Nacional de Investigação Agrária e Veterinária, Avenida da República, Quinta do Marquês (edifício sede), 2780-
10 157 Oeiras, Portugal

*Correspondence to*: Mohammad Farzamian (mohammad.farzamian@iniav.pt)

**Abstract**

Lezíria Grande of Vila Franca de Xira, located in Portugal, is an important agricultural system where soil faces the risk of salinization due to climate change, as the level and salinity of groundwater are likely to increase, as a result of the rise of the sea water level and consequently of the estuary. These changes can also affect the salinity of the irrigation water which is collected upstream of the estuary. Soil salinity can be assessed over large areas by the following rationale: (1) use of electromagnetic induction (EMI) to measure the soil apparent electrical conductivity ($EC_a$, mS m$^{-1}$); (2) inversion of $EC_a$ to obtain electromagnetic conductivity images (EMCI) which provide the spatial distribution of the soil electrical conductivity ($\sigma$, mS m$^{-1}$); (3) calibration process consisting of a regression between $\sigma$ and the electrical conductivity of the saturated soil paste extract ($EC_e$, dS m$^{-1}$), used as a proxy for soil salinity; and (4) conversion of EMCI into salinity cross sections using the obtained calibration equation.

In this study, EMI surveys and soil sampling were carried out between May 2017 and October 2018 at four locations with different salinity levels across the study area of Lezíria de Vila Franca. A previously developed regional calibration was used for predicting $EC_e$ from EMCI. Using time-lapse EMCI data, this study aims (1) to evaluate the ability of the regional calibration to predict soil salinity, and (2) to perform a preliminary qualitative analysis of soil salinity dynamics in the study area. The validation analysis showed that $EC_e$ was predicted with a root mean square error (RMSE) of 3.14 dS m$^{-1}$ in a range of 52.35 dS m$^{-1}$, slightly overestimated ($-1.23$ dS m$^{-1}$), with a strong Lin's concordance correlation coefficient (CCC) of 0.94 and high linearity between measured and predicted data ($R^2 = 0.88$). It was also observed that the prediction ability of the regional calibration is more influenced by spatial variability of data than temporal variability of data. Soil salinity cross sections were generated for each date and location of data collection, revealing qualitative salinity fluctuations related to the input of salts and water either through irrigation, precipitation, and level and salinity of groundwater. Time-lapse EMCI is developing into a valid methodology for evaluating the risk of soil salinization, so it can further support the evaluation and adoption of proper agricultural management strategies, especially in irrigated areas, where continuous monitoring of soil salinity dynamics is required.

**1 Introduction**

Lezíria Grande de Vila Franca de Xira (hereafter called Lezíria de Vila Franca) is an important agricultural system of alluvial origin located by the estuary of river Tejo, northeast of Lisbon, Portugal (Fig. 1), where soil faces risk of salinization due to the marine origin of part of the sediments, tidal influence of the estuary, irrigation practices, and projected evolution of future climate with increasing temperature and decreasing precipitation. Traditional soil salinity investigations have been conducted in the study area using the electrical conductivity of a saturated soil paste extract ($EC_e$, dS m$^{-1}$) as a proxy for soil salinity. However, they were limited to few boreholes and involved soil sampling, which restricted the analysis to point information, often lacking representativeness at the field scale. In addition, borehole drilling is invasive and not feasible to conduct over large areas, given the large number of boreholes that needs to be made.

Electromagnetic induction (EMI) is widely used as a non-invasive and cost-effective solution to map soil properties over large areas. EMI measures the apparent electrical conductivity of the soil ($EC_a$, mS m$^{-1}$), which is a function of soil properties such as salinity, texture, cation exchange capacity, water content and temperature. However, in a saline soil, soil salinity is generally the dominant factor responsible for the spatiotemporal variability of soil $EC_a$ when soil is moist. EMI surveys have been successfully used in conjunction with soil sampling to assess soil salinity through location-specific calibration between measured $EC_a$ and soil salinity (e.g. Triantafilis et al., 2000; 2001; Corwin and Lesch, 2005; Bouksila et al., 2012; Corwin and Scudiero, 2019; Kaufmann et al. 2019; von Hebel et al. 2019). However, the ability of this method for mapping soil salinity distribution with depth is limited. This is because EMI measures $EC_a$, a depth-weighted average conductivity measurement, which does not represent the soil electrical conductivity ($\sigma$, mS m$^{-1}$) with depth. More recently, a state-of-the-art approach called electromagnetic conductivity imaging (EMCI) has permitted to obtain $\sigma$ from the inversion of multi-height and/or multi sensor $EC_a$ data (Monteiro Santos, 2004; Dafflon et al., 2013; von Hebel et al., 2014; Farzamian et al., 2015; Shanahan et al., 2015; Jadoon et al, 2015; Moghadas et al., 2017). When comparing $\sigma$ with the soil properties sampled in boreholes, such as $EC_e$, soil water content, pH, among others, a calibration process is developed through a regression between $\sigma$ and the soil properties. This way, EMCI can be converted to a cross section of the soil properties which show strong correlation with $\sigma$. This methodology has been applied in Lezíria de Vila Franca to study soil salinity risk (Farzamian et al., 2019; Paz et al., 2019b), and salinity and sodicity risk (Paz et al., 2019a) in which EMCI has been converted to $EC_e$ and sodium adsorption

ratio. In this later study, the authors performed a principal component analysis of the soil properties in the study area, and found that the water content was correlated with sigma, but with a relatively lower influence when compared to the properties related to salinity ($EC_e$, SAR and ESP).

Because the inversion of $EC_a$ is relatively recent since the use of EMI for soil characterization, the lack of validation using an independent data set still limits the use of this new methodology (Corwin and Scudiero, 2019), making it therefore important to further test its accuracy in salinity monitoring.

When repeated over a period of time at the same place, EMCI becomes time-lapse EMCI and can be used to investigate the dynamics of soil properties such as soil water content (Huang et al., 2017; 2018; Moghadas et al., 2017). Using time-lapse

EMCI data, this study aims (1) to evaluate the ability of a previously developed regional calibration to predict soil salinity, and (2) to perform a preliminary qualitative analysis of soil salinity dynamics in the study area. For this purpose, EMI measurements and soil sampling were carried out between May 2017 and October 2018 at four locations with different salinity levels across the study area. EMI measurements were performed with a single-coil instrument (EM38), collecting $EC_a$ data in the horizontal and vertical orientations and at two heights, and then inverted to obtain EMCI, which provides a vertical

distribution of $\sigma$. Finally, $\sigma$ was converted to $EC_e$ using the previously developed regional calibration. Soil samples were collected along the EMI transects, and used for laboratory determination of $EC_e$. These data were used as an independent dataset to evaluate the ability of the regional calibration to predict soil salinity, and to generate soil salinity cross sections for each date of data collection.

## 2 Material and methods

### 2.1 Study area

The study was carried out in Lezíria de Vila Franca, a peninsula of alluvial origin surrounded by the rivers Tejo and Sorraia, and the Tejo estuary, located 10 km northeast of Lisbon, Portugal, as shown in Fig. 1. Soils in this region have fine to very fine texture and are classified as Fluvisols in the northern part and as Solonchaks in the southern part, according to the Harmonized World Soil Database (Fischer et al., 2012). Climate is temperate with hot and dry summers, according to the Köppen classification. Daily measurements of precipitation, mean temperature and reference evapotranspiration recorded during the study period at the meteorological station represented by the blue circle in Fig. 1b, are shown in Fig. 2. Land use in this area (of about 130 km$^2$) is constituted by irrigated annual crops in the northern part and mainly by rainfed pastures in the southern part. Irrigation is assured by an infrastructure that covers most of the area, collecting surface water at the confluence of the two rivers. The irrigation water has low salinity with electrical conductivity typically below 0.5 dS m$^{-1}$ and sodium adsorption ratio below 1 (mmol$_c$ L$^{-1}$)$^{0.5}$. The area exhibits a north-south soil salinity gradient which influences the distribution of land use types and which is probably due to the regional distribution of the marine fraction of sediments and to the saline influence of the estuary on groundwater in the southern part.

Four locations were chosen in the study area, as presented in Fig. 1b, with numbers 1 to 4. Locations 1, 2, and 3 are cultivated with annual rotations of irrigated herbaceous crops in spring and annual ryegrass (*Lolium multiflorum*) in the autumn, with ploughing usually once a year. During the study years (2017 and 2018), the spring crop at location 1 was tomato drip irrigated, and at locations 2 and 3 was maize irrigated by centre pivots. Location 4 is a rainfed spontaneous pasture that hasn't been ploughed at least in the last ten years. During the study period, location 1 was irrigated from 12 April to 23 July 2017 and from 30 May to 23 September 2018; location 2 was irrigated from 17 June to 11 October 2017 and from 24 May to 22 September 2018; and location 3 was irrigated from 17 May to 10 September 2017 and from 06 June to 17 September 2018. Groundwater level is shallow, as expected in an estuarine environment, and has saline characteristics. In the southern part of the study area, closer to the estuary, the depth and salinity of groundwater are influenced by tidal variation.

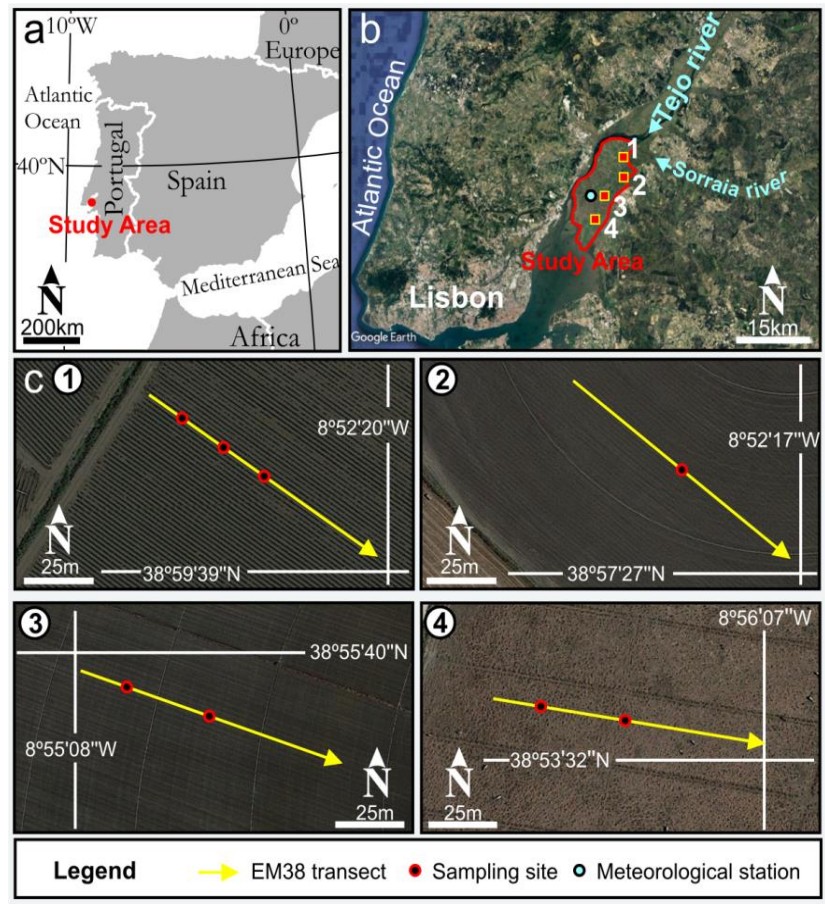

**Figure 1: (a–b) Location of the study area in Portugal, showing the main geographical features and the four locations; (c) details of the four locations showing the EM38 transects and the soil sampling sites © Google Earth.**

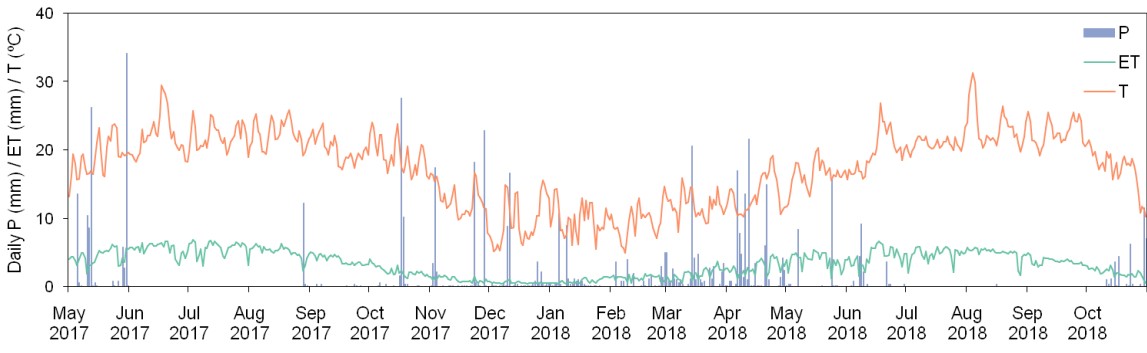

**Figure 2: Distribution of daily precipitation (P), reference evapotranspiration (ET) and mean temperature (T) recorded at the meteorological station located in the study area during the study period.**

## 2.2 Electromagnetic induction data acquisition and inversion

EMI data was acquired using the EM38 instrument (Geonics Ltd, Mississauga, Canada). Technology of this instrument is based on two coils, one transmitting the electromagnetic signal, and the other receiving it, distanced 1 m apart from each other inside the instrument case. The position of these coils can be controlled by placing the instrument in a vertical position relative to the soil surface - horizontal dipole mode (the coils stand in the horizontal position), which provides a maximum depth of investigation of 1.5 m - or in a horizontal position relative to the soil surface - vertical dipole mode (the coils stand in the vertical position), which provides a maximum depth of investigation of 0.75 m. EM38 surveys were done on five dates at locations 1 and 4, and on six dates at locations 2 and 3, during the period of May 2017 to October 2018. Measurements on the two first dates were continuously acquired at each location, along a 100 m transect, using a GPS (Rikaline 6010, with 5 m position accuracy) for registration of the position. Subsequent EMI measurements were acquired at each location, along a 20 m transect. The middle point of each 20 m transect was coincident with the medium point of each previous 100 m transect. Measurements were acquired at positions 1 m apart along the 20 m transects (Fig. 1c), overlapping the medium section of the 100 m transects. $EC_a$ was collected at two heights from the soil surface (0.15 and 0.4 m) in the horizontal and vertical dipole orientations, which was assured by placing the EM38 on a cart built specifically for this purpose. The cart has two shelves to accommodate the instrument, one at 0.15 m from the soil surface, and the other at 0.40 m from the soil surface. Inversion of $EC_a$ data to obtain σ was carried out using a 1-D laterally constrained inversion algorithm (Monteiro Santos et al., 2011). The $EC_a$ responses of the model were calculated through forward modelling based on the full solution of the Maxwell equations (Kaufman and Keller, 1983). The subsurface model used in the inversion process consisted of a set of 1-D models distributed according to the position of the $EC_a$ measurements. The subsurface model at each measurement position was constrained by the neighbouring models, allowing the use of the algorithm in regions characterized by high conductivity contrast. An Occam regularization (De Groot-Hedlin and Constable, 1990) based approach was used to invert the $EC_a$ data. All $EC_a$ data, collected at the four locations, were inverted by applying a five-layer earth initial model with electrical conductivity of 100 mS m$^{-1}$ and a fixed layer thickness of 0.30 m. To run the algorithm, several parameters were selected, such as the type of inversion

algorithm, the number of iterations, and the smoothing factor (λ) that controls the roughness of the model. The optimal inversion parameters for the present conditions were obtained in previous studies for the study area (Farzamian et al., 2019).

### 2.3 Soil sampling and laboratory analysis

Soil samples were collected at the same time of EMI surveys along the transects, as shown in Fig. 1c. At each sampling site, five soil samples were collected at 0.3 m increments, from a depth of 0.15 m to 1.35 m, as a representation of topsoil (0−0.3 m), subsurface (0.3−0.6 m), upper subsoil (0.6−0.9 m), intermediate subsoil (0.9−1.2 m), and lower subsoil (1.2−1.5 m), to monitor water content and $EC_e$. In the laboratory, water content was obtained using the gravimetric method, and then converted to volumetric water content ($\theta$ – $m^3\ m^{-3}$) after bulk density (g $m^{-3}$) determination from undisturbed 100 $cm^3$ soil samples. $EC_e$ was measured with a conductivity meter (WTW 1C20-0211 inoLab) in the extract collected from the soil saturation paste obtained from 300 g of air-dry soil samples, according to the methods described by Richards (1954). In this study, the soil is classified according to its $EC_e$ level as non-saline ($EC_e < 2$ dS $m^{-1}$), slightly-saline (2–4 dS $m^{-1}$), moderately-saline (4–8 dS $m^{-1}$), highly-saline (8–16 dS $m^{-1}$), and severely saline (>16 dS $m^{-1}$), according to the terminology proposed by Barrett-Lennard et al. (2008).

### 2.4 Prediction of $EC_e$ from time-lapse EMCI

A regional calibration to predict $EC_e$ from σ was previously developed for the study area resulting in the linear equation $EC_e = 0.03\sigma – 1.05$ (Farzamian et al., 2019). This calibration was termed "regional" because the equation was obtained using all $EC_e$ and σ data collected at four locations in the study area. Farzamian et al. (2019) tested the regional and location-specific calibrations, verifying that they have comparable prediction ability. However, the regional calibration can be used at any new location in the study area, within the range of measured $EC_e$, which makes it highly suitable for mapping salinity in the study area. The regional calibration was based on data collected during May and June 2017 and was validated using a leave-one-out-cross-validation method with good results (RMSE = 2.54 dS $m^{-1}$ in the 0–37 dS $m^{-1}$ range). The detailed calibration and cross-validation procedures are described in Farzamian et al. (2019).

In the present study, the regional calibration was used to predict $EC_e$ from time-lapse EMCI ($pEC_e$). The predicted $EC_e$ and $EC_e$ measured from soil samples ($mEC_e$), collected at the same time as the EMI surveys, were used as an independent data set

for the validation of the regional calibration. The validation was performed by calculating the root mean square error (RMSE), the coefficient of determination ($R^2$) between the measured and predicted $EC_e$, the Lin's concordance correlation coefficient (CCC), and the mean error (ME). Description of these statistical indicators and the equations used to calculate them are shown in Table 1. Calculations were done using global data, and also using data discriminated by date of measurement (in this case we considered dates when measurements were done at the four locations – January, June and October 2018).

**Table 1 – Description and equations of the statistical indicators used to evaluate the prediction ability of the regional calibration in this work.**

| Statistics | Equation[1] | | Description |
|---|---|---|---|
| Root mean square error (RMSE) | $$RMSE = \sqrt{\frac{\sum_{i=1}^{n}(mEC_{e_i} - pEC_{e_i})^2}{n-2}}$$ | (1) | Evaluates matching between measured and predicted data. When it is zero, it indicates perfect matching between measured and predicted data. |
| Mean error (ME) | $$ME = \frac{\sum_{i=1}^{n}(mEC_{e_i} - pEC_{e_i})}{n}$$ | (2) | Evaluates whether the predicted data are over- or underestimated. A negative value means overestimation, a positive value means underestimation. |
| Lin's concordance correlation coefficient (Lin's CCC) | $$Lin's\ CCC = \frac{2s_{mEC_e - pEC_e}}{s^2_{mEC_e} + s^2_{pEC_e} + (\overline{mEC_e} - \overline{pEC_e})^2}$$ | (3) | Evaluates agreement between measured and predicted data. Ranges from -1 to 1. When it is 1, it indicates perfect agreement between measured and predicted data (Lin, 1989). |

| Coefficient of determination ($R^2$) | $$R^2 = 1 - \frac{\sum_{i=1}^{n}(mEC_{e_i} - pEC_{e_i})^2}{\sum_{i=1}^{n}(mEC_{e_i} - \overline{mEC_e})^2}$$ | (4) | Indicates the proportion of the total variation of measured data that is explained by the calibration. Ranges from 0 to 1, although it may be negative values, which indicates an inappropriate calibration (Kvålseth, 1985). Above 0.5 is considered satisfactory. |

[1]$n$ is the total number of data; $mEC_e$ is measured $EC_e$; $pEC_e$ is predicted $EC_e$; the upper bar represents the mean of the indicated data.

To further explore the spatial and temporal sensitivity of the regional calibration and to investigate whether the regional calibration can be used to assess soil salinity variability at different locations and depths, we calculated the RMSE between the deviations of $mEC_e$ ($dmEC_e$) and the deviations of $pEC_e$ ($dpEC_e$) for each soil depth and location. These deviations were calculated by using the following equations:

$$dmEC_{e_i} = mEC_{e_i} - \overline{mEC_e} \tag{5}$$

$$dpEC_{e_i} = pEC_{e_i} - \overline{pEC_e} \tag{6}$$

in which the means were calculated with the values referent to each date of measurement for each soil depth and location. RMSE was then calculated between $dmEC_e$ and $dpEC_e$ according to equation (1).

## 4 Results and discussion

### 4.1 Temporal variation of measured $\theta$ and $EC_e$

Figure 3 shows the variation of $\theta$ and $EC_e$ with time at the sampling site located in the middle of each transect (Fig. 1c), at locations 1 to 4. At location 1, $\theta$ increases with depth and the lower subsoil (1.2−1.5 m) is permanently saturated within the study period. In the more superficial layers until 0.9 m depth, the influence of rainfall, evapotranspiration, and irrigation is noticeable. For instance, in the topsoil, $\theta$ peaks in January 2018 and lowers during the dry seasons, because drip irrigation during the dry seasons has a localized effect and there is high water uptake by the crop. At location 2, unlike the other locations, the lower subsoil is unsaturated. The influence of rainfall, evapotranspiration and irrigation is also noticeable. At locations 3 and 4, $\theta$ also increases with depth and the intermediate and lower subsoil layers are permanently saturated.

Regarding $EC_e$, at location 1 the values observed are always below 1 dS m$^{-1}$, except for the topsoil in September and October 2018, which is probably due to fertigation practises during the irrigation period. At location 2, $EC_e$ generally increases with depth. All layers show a peak in June and July 2018, probably due to fertigation practises. At location 3, $EC_e$ reaches higher levels than at the previous locations, exceeding 4 dS m$^{-1}$, which is the generally accepted threshold for the classification of saline soils. Location 4 presents the highest $EC_e$ of all locations. At the topsoil the values are below 4 dS m$^{-1}$, but increase consistently with depth to about 50 dS m$^{-1}$ in the lower subsoil. The increase of $EC_e$ during June 2018 can be due to the influence of saline groundwater.

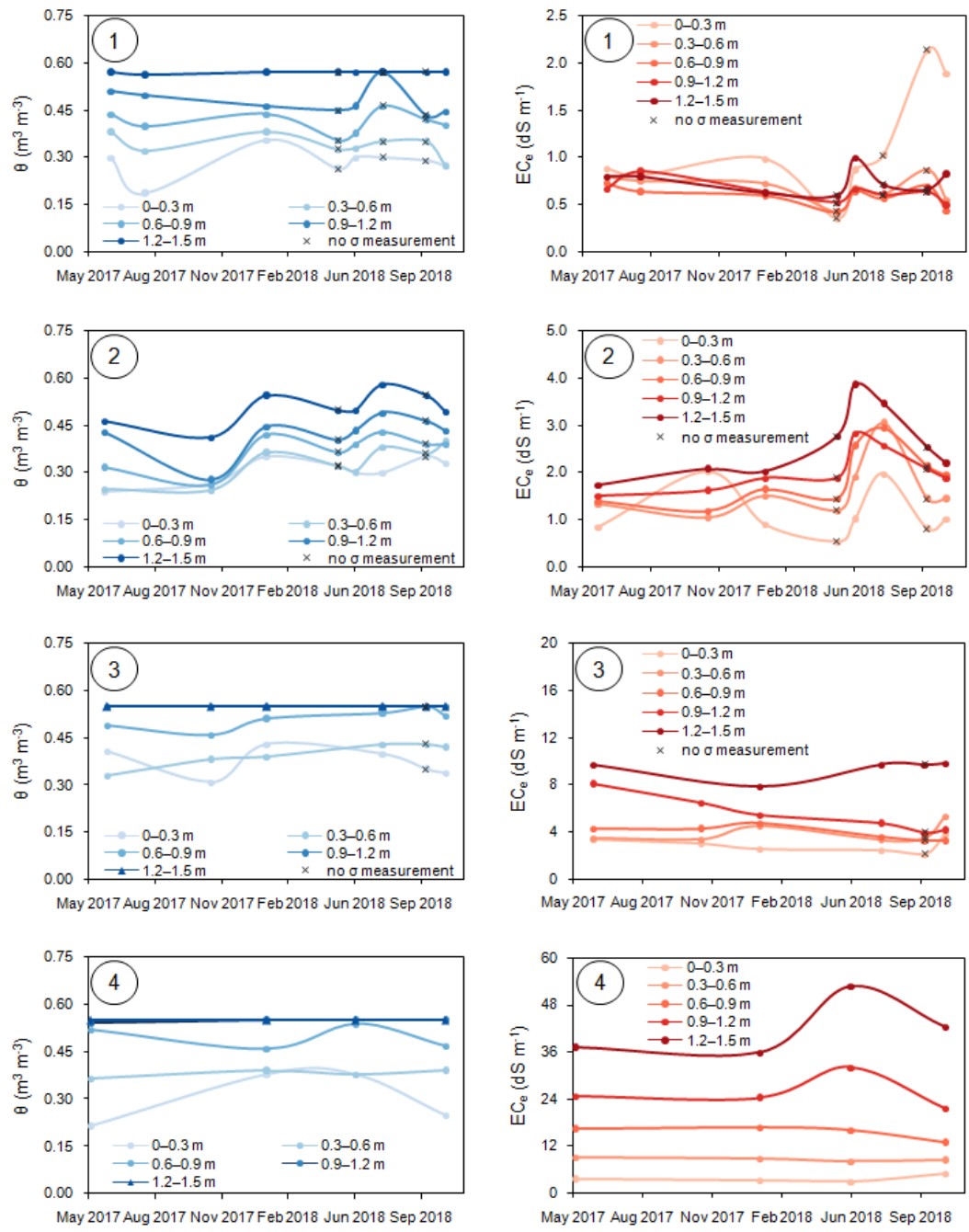

Figure 3: Volumetric water content ($\theta$ – m³ m⁻³) and electrical conductivity of the soil saturation extract (EC$_e$ – dS m⁻¹), in the topsoil (0–0.3 m), subsurface (0.3–0.6 m), upper subsoil (0.6–0.9 m), intermediate subsoil (0.9–1.2 m), and lower subsoil (1.2–1.5 m), measured in the sampling site located at the middle of each transect, at locations 1 to 4, during the study period. Each circled number refers to each location. Crosses refer to the dates when there were EC$_e$ measurements but no σ measurements, due to adverse field 205 conditions.

## 4.2. Time-lapse EMCIs

Figure 4 shows the obtained EMCIs at locations 1 to 4 for each date of the EMI surveys. Globally, $\sigma$ ranges from 19.44 mS m$^{-1}$ to 1431.57 mS m$^{-1}$ with the lowest values at location 1 and the highest at location 4. A general increasing trend of $\sigma$ is quite evident from the north to the south, accompanying the previously known soil salinity gradient. In addition, $\sigma$ increases with depth at locations 2, 3 and 4. At location 1, $\sigma$ ranges spatiotemporally from 19.44 mS m$^{-1}$ to 128.08 mS m$^{-1}$. At location 2, $\sigma$ ranges from 28.02 mS m$^{-1}$ to 469.39 mS m$^{-1}$ with highest values at depth. A similar pattern of $\sigma$ is evident at locations 3 and 4. However, a greater range of $\sigma$ is seen at location 3 with values from 36.23 mS m$^{-1}$ to 706.32 mS m$^{-1}$. Location 4 exhibits the largest variations of $\sigma$, ranging from 48.57 mS m$^{-1}$ to 1431.57 mS m$^{-1}$.

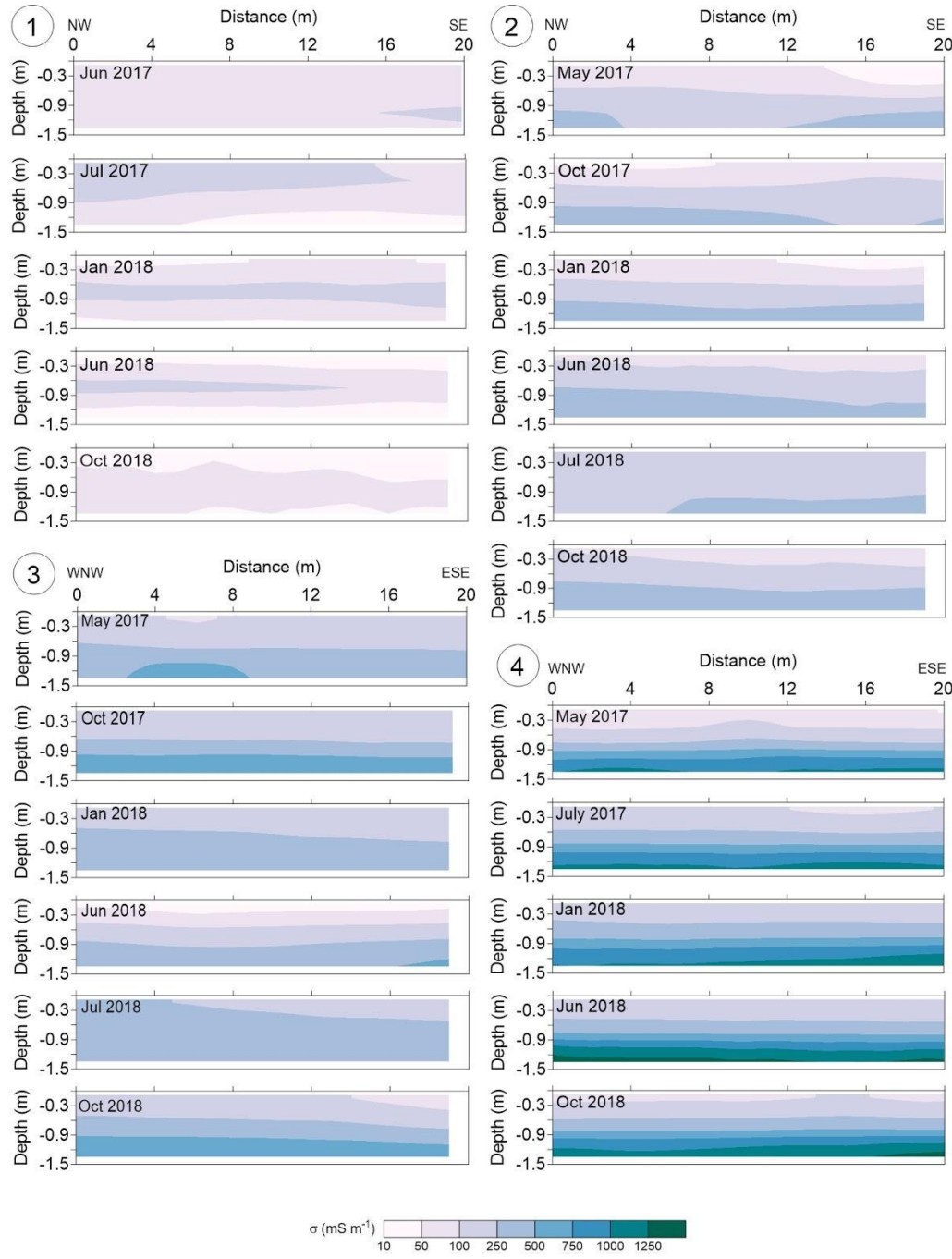

**Figure 4: Time-lapse electromagnetic conductivity images (EMCIs) for locations 1 to 4.**

### 4.3 Prediction of $EC_e$ using the regional calibration

Figure 5 shows $EC_e$ predicted with the regional calibration versus the measured $EC_e$ and the 1:1 line, with points identified in terms of date of measurement (Fig. 5a) and depth of measurement (Fig. 5b). Table 2 shows the statistical indicators obtained
using global data, i.e., data collected at all locations, from July 2017 to October 2018, and the statistical indicators for each date. The validation of the regional calibration using global data resulted in a RMSE of 3.14 dS m$^{-1}$ and a R$^2$ of 0.88, which indicates satisfactory prediction ability, given the large range of $EC_e$ (52.35 dS m$^{-1}$). The high global Lin's CCC of 0.94 shows agreement between measured and predicted $EC_e$. The ME is $-1.23$ dS m$^{-1}$, indicating that the regional calibration generally overestimates $EC_e$. Figure 5a and Fig. 5b show that the points are generally scattered around the 1:1 line and it is not possible
to identify variations depending on the date or depth of the measurement. In order to analyze the prediction ability at each location, Fig. 5c and Fig. 5d display an enlargement of the lower left part of the previous figures, displaying $EC_e$ values below 15 dS m$^{-1}$. Figure 5c and Fig. 5d show differences in the prediction ability according to the location, namely at locations 2 and 3, where $EC_e$ is generally overestimated. At location 2, $EC_e$ is more overestimated in deeper soil layers (Fig. 5d) which is likely due to the clay content that consistently increases with depth at this location, while it is rather uniform or declines with depth
at the other locations (Farzamian et al., 2019). At location 3, $EC_e$ is also overestimated, most likely due to the influence of $\theta$ and cation exchange capacity (Paz et al., 2019a) which are higher on average compared to locations 2 and 4. Finally, the $EC_e$ ranges of location 4 and of the lower subsoil are similar to the $EC_e$ range of global data, showing dominance of location 4 and of lower subsoil data on the calibration. On the other hand, the statistical indicators discriminated by date of measurement (in this case we considered only the dates when measurements were done at the four locations – January, June and October 2018),
shown in Table 2, reveal that the prediction ability doesn't vary significantly when comparing the statistical indicators of the three dates. These results suggest that spatial variability of data has a much stronger influence on the prediction ability of the regional calibration, than temporal variability of data.

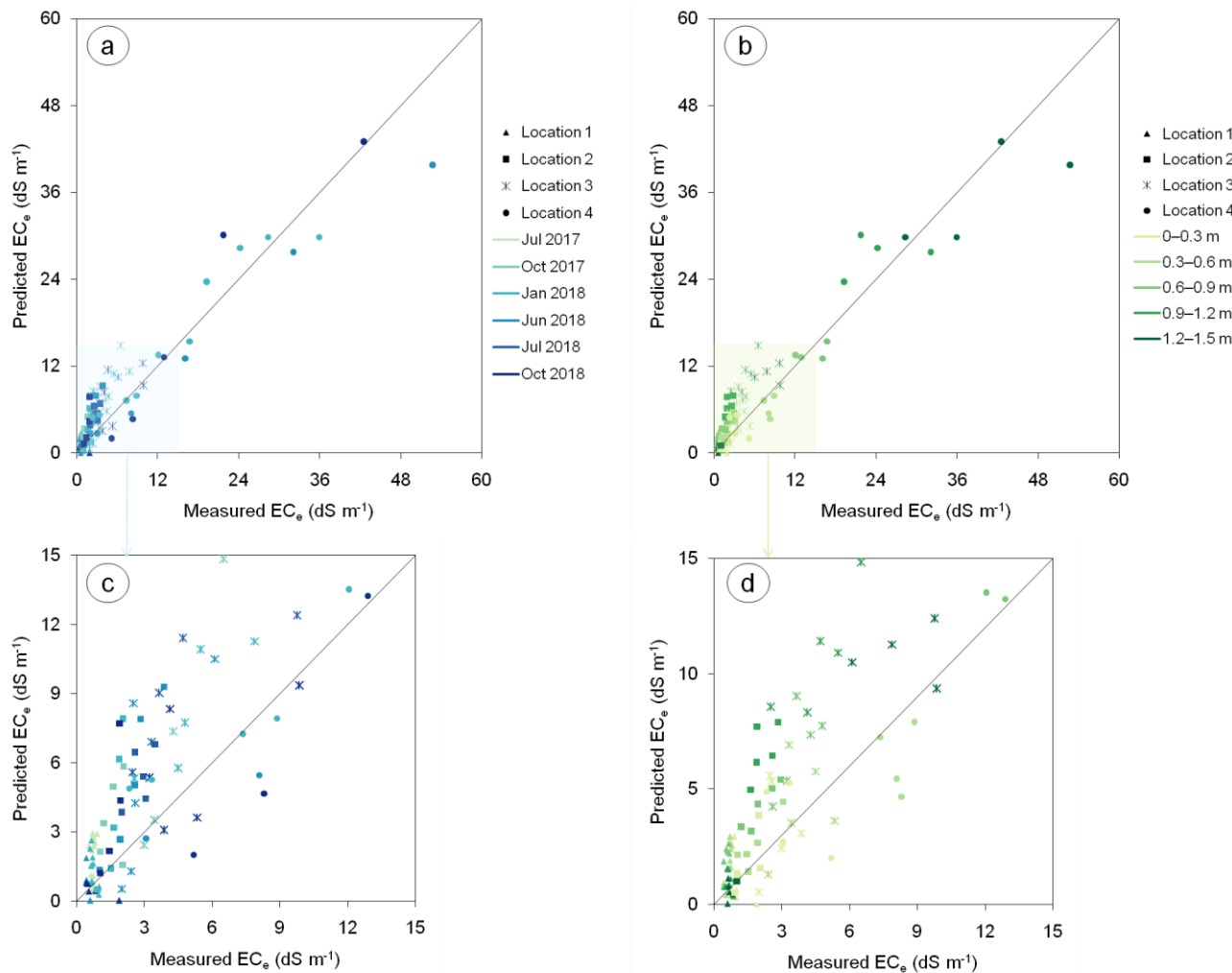

**Figure 5: Plots of predicted EC$_e$ versus measured EC$_e$ and the 1:1 line, obtained for locations 1 to 4, identified in terms of date of measurement (a) and depth of measurement (b). Plots (c) and (d) show enlargements of the lower left part of plots (a) and (b), respectively.**

**Table 2 – RMSE, ME, Lin's CCC, R2, minimum, maximum and range of ECe, and the number of data used to calculate these statistical indicators, discriminated in terms of global, and date of measurement.**

| | RMSE (dS $m^{-1}$) | ME (dS $m^{-1}$) | Lin's CCC | $R^2$ | $EC_e$ min (dS $m^{-1}$) | $EC_e$ max (dS $m^{-1}$) | $EC_e$ range (dS $m^{-1}$) | Number of data |
|---|---|---|---|---|---|---|---|---|
| Global | 3.14 | −1.23 | 0.94 | 0.88 | 0.35 | 52.70 | 52.35 | 103 |
| Jan 2018 | 2.79 | −1.33 | 0.96 | 0.91 | 0.59 | 35.90 | 35.31 | 30 |
| Jun 2018 | 4.27 | −0.08 | 0.94 | 0.90 | 0.35 | 52.70 | 52.35 | 20 |
| Oct 2018 | 3.11 | −0.71 | 0.96 | 0.91 | 0.44 | 42.50 | 42.06 | 19 |

Table 3 shows the RMSE between dmEC$_e$ and dpEC$_e$ obtained for each soil depth and each location. The RMSE is 1.04 dS $m^{-1}$ at topsoil, which is relatively high given the small range of deviations variability (2.18 dS $m^{-1}$). This indicates that it is difficult to estimate soil salinity variability at topsoil from time-lapse EMI data and regional calibration. We attribute the weak prediction ability at topsoil to the small range of EC$_e$ variability and to the larger variability of other soil properties (e.g. θ and temperature) at topsoil, which are due to different irrigation schemes and cultivated crops at each location. The results of RMSE show better prediction ability at subsurface and very good prediction ability at upper, intermediate and lower subsoil suggesting that the time-lapse EMI data and regional calibration can be used to assess soil salinity changes in these soil layers. On the other hand, the RMSE results for each location show that the prediction ability is poor for monitoring of soil salinity variability at location 1 (1.06 dS $m^{-1}$) and location 2 (2.00 dS $m^{-1}$) given the small range of deviations variability at these locations. This is not surprising as the soil is non-saline at location 1 with a very small range of EC$_e$ (0.35–1.89 dS $m^{-1}$) and slightly-saline at location 2 with a small range of EC$_e$ (0.91–3.86 dS $m^{-1}$). Thus, other soil properties such as θ and clay content have larger impact on spatiotemporal variability of σ. In addition, larger variations of EC$_e$ and σ at location 4 dominated the regional regression calibration, further limiting the application of the regional calibration for monitoring the small variability of soil salinity at location 1 and 2. The RMSE results are better to some extent at location 3 and very good at location 4 suggesting that the time-lapse EMI data and regional calibration can fairly predict soil salinity variability at location 3 and very well at location 4.

This spatial sensitivity of the regional calibration can be improved by studying new locations across the study area to include a wider variability of soil properties and ranges of EC$_e$ in the regression calibration. On the other hand, longer observation periods and more frequent EMI surveying and soil sampling, as well as monitoring of other soil dynamic properties that

influence σ (i.e. θ, soil temperature, level and salinity of groundwater), and finding ways to quantitatively account for their impact on time-lapse EMCIs, can improve the temporal sensitivity of regional calibrations.

**Table 3 – RMSE between dmEC$_e$ and dpEC$_e$, range of dmEC$_e$, and the number of data used to calculate RMSE, discriminated in terms of depth of measurement and location.**

| | RMSE (dS m$^{-1}$) between dmEC$_e$ and dpEC$_e$ | dmEC$_e$ range (dS m$^{-1}$) | Number of data |
|---|---|---|---|
| 0–0.3 m | 1.04 | 2.18 | 21 |
| 0.3–0.6 m | 1.14 | 4.91 | 21 |
| 0.6–0.9 m | 1.38 | 10.63 | 21 |
| 0.9–1.2 m | 2.76 | 22.61 | 21 |
| 1.2–1.5 m | 4.18 | 36.88 | 19 |
| Location 1 | 1.06 | 1.45 | 35 |
| Location 2 | 2.00 | 2.85 | 24 |
| Location 3 | 2.50 | 7.47 | 24 |
| Location 4 | 3.88 | 49.64 | 20 |

**4.4 Generation of soil salinity cross sections from time-lapse EMCI**

Figure 6 shows the soil salinity cross sections (EC$_e$ predicted using the regional calibration) at locations 1 to 4 for each date of the EMI surveys, categorized into 6 salinity classes, ranging from non-saline to severely-saline. The measured EC$_e$ and the groundwater level at the sampling site located in the middle of each EMI transect are also shown.

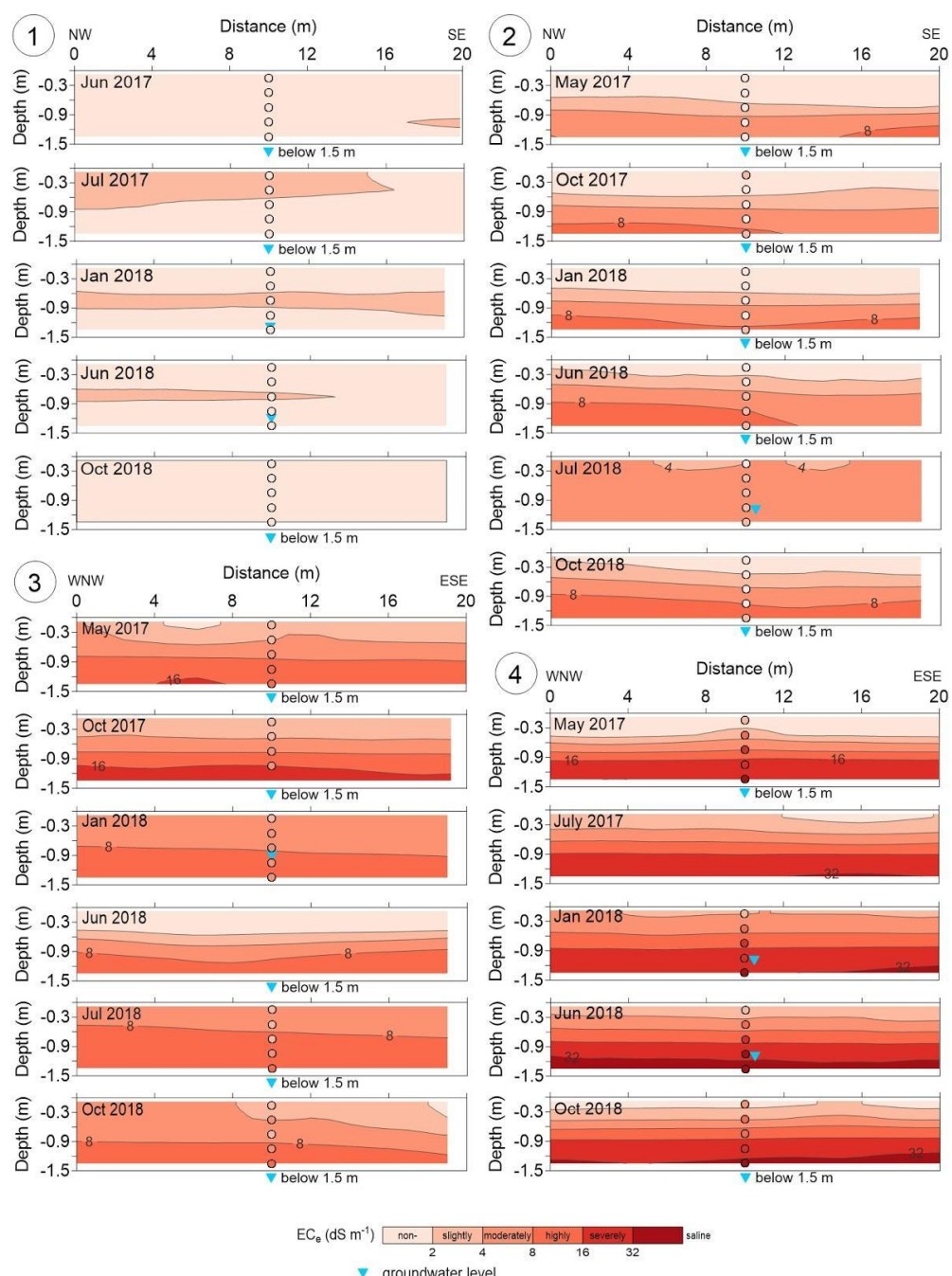

**Figure 6: Cross sections of soil salinity (predicted $EC_e$) for locations 1 to 4, with representation of measured $EC_e$ (in circles) and groundwater level (blue triangles) at the sampling sites located in the middle of each transect. Note that in June 2018 at location 3 and in July 2017 at location 4 there was no soil sampling.**

The salinity cross sections for location 1 show that the soil is generally non-saline, with slightly saline zones in all dates except for October 2018. These saline zones occur in the top soil layers until 0.9 m depth (topsoil, subsurface and upper subsoil), and represent an overestimation of the soil salinity when compared to the measured $EC_e$ of the sampling point (which is invariably non-saline). This overestimation tendency is in agreement with Fig. 5d, where the very low range of spatiotemporal variations of soil salinity at this location can also be observed. In such conditions, other soil properties, such as $\theta$, dominate the small variations of $\sigma$, and therefore the ability to predict salinity from $\sigma$ at this location was reduced as discussed in section 4.3. Our previous studies with both location-specific and regional calibrations tested at this location showed similar results (Farzamian et al., 2019).

At location 2 the salinity cross sections show an increase of salinity with depth from non-saline at the topsoil to highly-saline in the lower subsoil, with exception of July 2018, where the entire soil profile is moderately saline. The increase of soil salinity in upper soil layers in July 2018 can be attributed to fertigation practices for the maize cultivation that introduced salts into the soil profile. The salinity cross sections also show the overestimation of salinity occurring mainly at deeper soil layers, which agrees with the results presented in Fig. 5d and discussed in section 4.3.

At location 3 soil salinity is well predicted in May 2017 but tends to be slightly overestimated in the remaining dates, especially in July 2018. The salinity cross sections show that salinity increases with depth reaching severely-saline in May 2017 and October 2017. This can be due to the influence of the saline groundwater (as seen in Fig. 3, the intermediate and lower subsoil layers are permanently saturated). The groundwater level is above 1.5 m in January 2018, although the salinity of the deeper soil layers (>0.9 m) decreases compared to May and October 2017, which could be due to washing of the profile by rainfall. The increase of soil salinity in upper soil layers in July 2018, similarly to location 2 on the same date, can be attributed to fertigation practices for the maize cultivation.

At location 4 the trend of increasing salinity with depth is accurate in all dates, but it tends to be slightly underestimated. The salinity cross sections show that salinity increases from non-saline in topsoil to severely-saline in lower subsoil. This is probably related to the saline groundwater level above 1.5 m. During the dry period of the year, salinity of the lower subsoil reaches the highest values (June 2018).

Comparison of the salinity cross sections between locations confirms the previously known north-south soil salinity spatial gradient of the study area, that is, from location 1 to 4, soil salinity generally increases. Soil salinity dynamics at each location reveals fluctuations in time related to the input of salts and water either through irrigation, precipitation or groundwater level and salinity. Location 1 tends to have non-saline characteristics, which can be attributed to good quality irrigation. In addition, this location is far from the estuary, making it less prone to the presence of saline groundwater. At locations 2 and 3, the salinity cross sections show an increase of soil salinity in the upper layers during the dry season (when irrigation occurs), which decreases in the following months with increased rainfall (Fig. 2). At location 4, an increment of salinity along the entire profile is visible during the dry season. This is likely due to the influence of the saline groundwater and capillary rise along the profile.

## 5 Conclusions

In this study, EMI and soil sampling data collected between May 2017 and October 2018 were used, together with a previously developed regional calibration, to predict soil salinity. This procedure allowed further validation of the regional calibration with an independent dataset and a preliminary qualitative analysis of soil salinity dynamics in the study area. Based on the comprehensive analysis of the statistical indicators obtained from the validation process, and the obtained soil salinity cross sections, the following main conclusions can be drawn:

1. The validation performed in this study resulted in a RMSE of 3.14 dS m$^{-1}$, which is acceptable given the large range of $EC_e$ (52.35 dS m$^{-1}$). This validation resulted in lower prediction ability than that previously resulting from cross-validation (which had a RMSE of 2.54 dS m$^{-1}$). This is expected because the test set is completely independent from the dataset used to develop the calibration and was collected over a wider period of time (18 months). During this period, soil properties, which are also known to influence $\sigma$, such as temperature and $\theta$, change, which introduces larger variability in data.

2. The prediction ability of the regional calibration does not vary significantly over time. As a result, the regional calibration approach still stands as an expeditious method to predict soil salinity from EMI surveys at any new location in the study area. However, prediction ability of the regional calibration in assessing variability of soil salinity at different depths and locations varies significantly due to variability of soil properties at each depth and location. Our investigation shows that significantly larger variations of $EC_e$ and $\sigma$ at location 4 dominated the regional regression calibration, suggesting a good prediction ability of the regional calibration in the south of the study area and close to location 4 where the soil salinization is of major concern and can compromise agricultural activity.

3. The methodology used in this study allowed the generation of soil salinity cross sections displaying the patterns of soil salinity at different dates, at four locations in the study area. The salinity cross sections show a qualitative response of soil salinity to the input of salts and water either through irrigation, precipitation or level and salinity of groundwater. In a regional perspective, soil salinity dynamics in the study area may be preliminarily explained by a combination of spatial distribution of the marine fraction of soil, with irrigation practices in the study area and saline groundwater in the southern part.

Application of time-lapse EMCI and calibration for assessing soil salinity dynamics is a developing methodology that can further support the evaluation and adoption of proper agricultural management strategies in irrigated regions. Some aspects can and will be addressed in future studies so as to improve its performance. From this study, we identify some of these aspects. First, relatively to the inversion process, and in the absence of a time-lapse inversion algorithm, $EC_a$ data was inverted independently. This method can distort the inversion results, since the reference model and a priori information are not considered. Further research involves time-lapse inversion algorithms that are being developed to invert data collected with EMI sensors, which can generate EMCIs of higher precision. Secondly, the influence of static soil properties (i.e., that do not vary in time), such as clay content and cation exchange capacity, could be tackled with the use of cross sections of the variation of soil salinity between two consecutive dates, which allows removing the static effect from the time-lapse EMCIs. Finally, temporal soil salinity assessment can be optimized by quantitatively taking into account the influence of soil dynamic properties on the time-lapse EMCIs. Specifically, in Lezíria, regional calibrations can be improved by studying new locations across the study area for a longer period of time with more frequent surveying and sampling, and also by including new parameters, such as $\theta$, soil temperature, level and salinity of groundwater. However, the temporal variations of these properties are connected to location specific conditions. For instance, $\theta$ can vary significantly in the study area, particularly in the root zone, due to different irrigation practices, root uptake of different crops, and fluctuation of groundwater level. These facts highlight the necessity of using location-specific calibrations for a more precise assessment of soil salinity changes at each location.

*Data availability.* The data were acquired at privately owned farms and are not available to the public.

*Competing interests.* The authors declare that they have no conflict of interest.

*Author contributions.* MCP and MF designed EMI surveys; processed and inverted time-lapse EMI data; performed the statistical analysis and generated the main results. AMP, NLC and MCG collected soil samples and performed the laboratory analysis and contributed to the statistical analysis. FMS developed the code for inversion of EMI data. MCP, MF, AMP and NLC wrote the text. MCP prepared the visualization. MCG and FMS contributed to the elaboration of the methodology and discussion of the results.

**Acknowledgements**

The authors are grateful to the Associação de Beneficiários da Lezíria Grande de Vila Franca de Xira and to Manuel Fernandes and Fernando Pires from INIAV for field assistance.

This work was funded by the Portuguese research agency, Fundação para a Ciência e a Tecnologia (FCT), in the scope of

project SALTFREE – ARIMNET2/0004/2015 SALTFREE and ARIMNET2/0005/2015 SALTFREE. Publication is supported by FCT – project UID/GEO/50019/2019 – Instituto Dom Luiz.

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
