# Peer review of "Assessing soil salinity dynamics using time-lapse electromagnetic conductivity imaging"

_SOIL, 2019_

## Referee Comment (RC1) · Anonymous Referee #1 · 9 Feb 2020

In this paper, the authors explored the use of time-lapse EMI surveys and a 1-D inversion algorithm to predict soil salinity (ECe) at different depths at four sites in Lezíria de Vila Franca. A pre-established calibration equation was used to convert sigma to ECe. The model performance was good given the large R2, and Lin's concordance. They conclude that the method has the potential to be used for the rapid monitoring of soil salinity dynamics for agricultural management.

In general, the study is well-written and provides a nice example of monitoring soil salinity using non-invasive EMI surveys.

I have a few comments about the manuscript.

1. The authors need to elaborate more on the pre-calibration of the EM38 meter. How

are the variations of soil temperature (affecting soil ECa) be corrected?

2. Please discuss more on the pre-determined linear calibration equation. Since the soil water content is measured, can you include this in the model to improve the model performance? Can the variations in soil water content or soil texture explain the small misfit between the predicted ECe and measured ECe?

3. If possible, is it possible to include soil water content in the calibration model and develop a pedo-transfer physical model coupled with a novel inversion algorithm to simultaneously retrieve soil water and ECe values from time-lapse EMI surveys of ECa values in the future?

---

## Referee Comment (RC2) · Anonymous Referee #2 · 13 Mar 2020

[referee-annotated manuscript omitted]

---

## Author Comment (AC1) · 17 Apr 2020

**"Monitoring soil salinity using time-lapse electromagnetic conductivity imaging" - authors' responses to suggestions and comments made in Public Discussion**

Dear editor and referees,

We sincerely appreciate and thank your constructive comments on our manuscript. We have revised the manuscript according to your suggestions and comments. We hope that the revised version of the manuscript properly addresses your concerns.

Our answers are placed in blue below each of the referees' interactive comments, and also below each of Referee #2's comments and corrections, provided in the supplementary annotated .pdf. The changes to the manuscript, arising from the referee's comments, are written in grey. We also added a section with some additional changes.

Sincerely, Mohammad Farzamian on behalf of all authors

**Authors responses to referees' comments**

**Anonymous Referee #1**

https://doi.org/10.5194/soil-2019-99-RC1, 2020

In this paper, the authors explored the use of time-lapse EMI surveys and a 1-D inversion algorithm to predict soil salinity (ECe) at different depths at four sites in Lezíria de Vila Franca. A pre-established calibration equation was used to convert sigma to ECe. The model performance was good given the large R2, and Lin's concordance. They conclude that the method has the potential to be used for the rapid monitoring of soil salinity dynamics for agricultural management.

In general, the study is well-written and provides a nice example of monitoring soil salinity using non-invasive EMI surveys. I have a few comments about the manuscript.

We would like to thank Anonymous Referee #1 for evaluating our manuscript. We highly appreciate the overall positive comments.

1. The authors need to elaborate more on the pre-calibration of the EM38 meter. How are the variations of soil temperature (affecting soil ECa) be corrected?

The calibration of EM38 was done using the routine procedure (inphase nulling and instrument zero) recommended by the company at the beginning of measurements at each field.

We have also performed electrical resistivity tomography (ERT) surveys at the four locations and along the same EMI transects right after EMI surveys (2-3 times at each location), in order to evaluate by comparison the precision of the EMI data and models, as well as to explore the potential drift of the EM38 sensor. The ERT technique has usually a better precision in measuring soil electrical conductivity, but it is more effort- and time-consuming in the field, which limits its application for large-area investigations. The obtained ERT models agree with the EMCIs shown in this paper, which gives confidence to the EMCIs models. However, the results of this comparison are not in the scope of this paper.

Unfortunately, borehole temperature data were not available at the study sites (which are private farms) to correct the impact of soil temperature changes over the EM38 models. We agree that the lack of temperature correction will add to the uncertainty of temporal  $EC_e$  changes assessment, although the impact of temperature changes over soil electrical conductivity is expected to be relatively small compared to soil salinity. We revised the text including the introduction, results and conclusion to address this issue.

**Old version (line 244)**

"Furthermore, the influence of static soil properties (i.e., that do not vary in time), such as clay content, could be tackled with the use of maps of the variation of soil salinity between two consecutive dates, which allows removing the static effect in the EMCIs."

**New version**

"In addition, the influence of static soil properties (i.e., that do not vary in time), such as clay content, could be tackled with the use of cross sections of the variation of soil salinity between two consecutive dates, which allows removing the static effect from the EMCIs. Future attempts can be made on finding ways to quantitatively account for the impact of dynamic soil properties (i.e.  $\theta$  and temperature) on the EMCIs to optimize temporal soil salinity assessment."

**Old version (line 45)**

"EMI measures the apparent electrical conductivity of the soil ( $EC_a$ , dS m-1), which is primarily a function of soil salinity, soil texture, water content, and cation exchange capacity; however, in a saline soil, soil salinity is generally the dominant factor responsible for the spatiotemporal variability of soil  $EC_a$ ."

**New version**

"EMI measures the apparent electrical conductivity of the soil ( $EC_a$ , dS m-1), which is a function of soil properties such as salinity, texture, cation exchange capacity, water content and temperature. However, in a saline soil, soil salinity is generally the dominant factor responsible for the spatiotemporal variability of soil  $EC_a$ ."

**Old version (line 183)**

"Furthermore, this test set is composed of measurements collected over a wider period of time (18 months). During this period, soil properties, which are also known to influence  $\sigma$ , such as  $\theta$ , change (as shown in Fig. 3), which introduces larger variability in the measurements."

**New version**

"Furthermore, this test set is composed of measurements collected over a wider period of time (18 months). During this period, soil properties, which are also known to influence  $\sigma$ , such as temperature and  $\theta$ , change , which introduces larger variability in the measurements."

2. Please discuss more on the pre-determined linear calibration equation. Since the soil water content is measured, can you include this in the model to improve the model performance? Can the variations in soil water content or soil texture explain the small misfit between the predicted ECe and measured ECe?

The water content could be included along with  $\sigma$  in the model for prediction of ECe. As these variables are linearly correlated, this could be achieved through a principal component regression (PCR). But in this paper our aim was to test the validity of the regional calibration over time, which includes wide variations of water content and other conditions, because the strength of the regional calibration is that it is very practical for monitoring salinity in this large area. If the water content would be included as a predictor in the model, it would have to be measured along with  $\sigma$  in the field, which would increase the complexity of the monitoring process.

In previous studies we also found that, in this study area, the water content is correlated with  $\sigma$ , but that properties related to salinity (ECe, SAR and ESP) have a larger influence on  $\sigma$ . We propose to add the following text for further clarifying this aim and its limitations. We also strengthen the importance of validating the previous approach with an independent test set, adding a new paragraph.

**Old version (line 59)**

"(...) and sodium adsorption ratio."

**New version**

"(...) and sodium adsorption ratio. In this later study, the authors performed a principal component analysis of the soil properties in the study area, and found that the water content was correlated with sigma, but with a relatively lower influence when compared to the properties related to salinity ( $EC_e$ , SAR and ESP).

Because the inversion of  $EC_a$  is relatively recent since the use of EMI for soil characterization, the lack of validation using an independent data set still limits the use of this new methodology (Corwin and Scudiero, 2019), making it therefore important to further test its accuracy in salinity monitoring."

**The reference will also be added to the references list.**

"Corwin, D.L. and Scudiero, E.: Chapter One - Review of soil salinity assessment for agriculture across multiple scales using proximal and/or remote sensors. Adv Agron, 158, 1–130, https://doi.org/10.1016/bs.agron.2019.07.001, 2019."

The soil electrical conductivity is a complex function of soil properties, including soil texture, water content and salinity. Therefore, water content and soil texture could impact the misfit between the predicted  $EC_e$  and measured  $EC_e$ .

3. If possible, is it possible to include soil water content in the calibration model and develop a pedo-transfer physical model coupled with a novel inversion algorithm to simultaneously retrieve soil water and ECe values from time-lapse EMI surveys of ECa values in the future?

It is mathematically possible to develop such an inversion algorithm, but it goes quite beyond the scope of the present paper. It is worth mentioning that one can not estimate both water content and  $EC_e$  from soil electrical conductivity alone as both parameters influence soil electrical conductivity. On the other hand, including water content in the  $EC_e$  estimation and possible inversion algorithm requires water content measurements along with EMI measurements in the field. Please see also our answer to the previous question.

**Anonymous Referee #2**

https://doi.org/10.5194/soil-2019-99, 2020

Dear Maria Catarina Paz et al., the study entitled 'Monitoring soil salinity using time-lapse electromagnetic conductivity imaging' provides an overview of salinty assessment using EMI data and inversions together with ground-truth data in an agricultural area close to Lisboa. In general, the manuscript is very well written and the narrative is nicely presented. With this review, I enclosed some comments in the annotated .pdf ('soil-2019-99supplement.pdf') that is provided with the review process. Please consider the comments in that pdf for further improvements. Including my aspects together with the suggestions of the previous reviewer, I'd recommend publishing the manuscript in SOIL.

**Kind regards, Referee 2**

We would like to thank Anonymous Referee #2 for evaluating our manuscript. We highly appreciate the overall positive comments.

**Supplementary .pdf**

Line 24: "<del>satisfactorily"</del> OK. We have revised the text accordingly.

**Old version (line 24)**

"The results showed that ECe was satisfactorily predicted, (...)".

**New version:**

"The results showed that ECe was predicted, (...)".

**Line 47: salinity is dominant as long as soil is moist**

We agree and have revised the text accordingly.

**Old version (line 47)**

"(...) however, in saline soil, soil salinity is generally the dominant factor responsible for the spatiotemporal variability of soil  $EC_a$ ."

**New version**

"(...) however, in saline soil, soil salinity is generally the dominant factor responsible for the spatiotemporal variability of soil  $EC_a$  when soil is moist."

Line 54: please include here some more studies that use EMI and EMI inversions. e.g., Corwin&Scudiero (Advances in Agronomy), von Hebel et al. 2019 (sensors), Kaufmann et al. 2020 (Soil Use and management), Wang et al. 2019 (JGR), Guillemoteau et al. 2019 (GJI)

We have now included more references.

**Old version (line 47)**

"EMI surveys have been successfully used in conjunction with soil sampling to assess soil salinity through location-specific calibration between measured  $EC_a$  and soil salinity (e.g. Triantafilis et al., 2000; 2001; Corwin and Lesch, 2005; Bouksila et al., 2012)."

**New version**

"EMI surveys have been successfully used in conjunction with soil sampling to assess soil salinity through location-specific calibration between measured ECa and soil salinity (e.g. Triantafilis et al., 2000; 2001; Corwin and Lesch, 2005; Bouksila et al., 2012; Corwin and Scudiero, 2019; Guillemoteau et al., 2019; von Hebel et al. 2019; Wang et al., 2019; Kaufmann et al. 2020)."

Line 62: Please rephrase the sentence that no study maps salinity. Actually, this is why the EM38 was invented back in the 1980-ties. Numerous EMI studies map salinity. Also with inversion. E.g., by Triantafillis group.

We agree that many studies inverted EMI data for soil salinity mapping. However, the sentence relates to the inversion of **time-lapse** data for **monitoring** soil salinity in time which has not been reported yet to the best of our knowledge. No changes to the text were made relative to this comment.

Line 103: Although the EM38 is well known, please include the basic information like coil distance, coil orientation, depth range of investigation. This also help "new readers" guiding through the inversion results. Agreed. We have revised the text accordingly.

**Old version (line 103)**

"EMI data was acquired using the EM38 instrument (Geonics Ltd, Mississauga, Canada) on five dates at locations 1 and 4, and on six dates at locations 2 and 3, during the period of May 2017 to October 2018."

**New version**

"EMI data was acquired using the EM38 instrument (Geonics Ltd, Mississauga, Canada). Technology of this instrument is based on two coils, one transmitting the electromagnetic signal, and the other receiving it, distanced 1 m apart from each other inside the instrument case. The position of these coils can be controlled by placing the instrument in a vertical position relative to the soil surface - horizontal dipole mode (the coils stand in the horizontal position), which provides a maximum depth of investigation of 1.5 m - or in a horizontal position relative to the soil surface - vertical dipole mode (the coils stand in the vertical position), which provides a maximum depth of investigation of 0.75 m. EM38 surveys were done on five dates at locations 1 and 4, and on six dates at locations 2 and 3, during the period of May 2017 to October 2018."

**Line 105: accuracy of GPS?**

GPS has 5 m position accuracy. We can add the accuracy of the GPS to the sentence.

**Old version (line 104)**

"Measurements on the two first dates were continuously acquired along 100 m transects using a GPS (Rikaline 6010) for registration of the position."

**New version**

"Measurements on the two first dates were continuously acquired at each location, along a 100 m transect, using a GPS (Rikaline 6010, with 5 m position accuracy) for registration of the position."

**Line 106: unclear description of measurement procedure along/at the transects**

We have revised the text accordingly.

**Old version (line 105)**

"Subsequent EMI measurements were acquired at positions 1 m apart along 20 m transects (Fig. 1c), overlapping the medium section of the 100 m transects."

**New version**

"Subsequent EMI measurements were acquired at each location, along a 20 m transect. The middle point of each 20 m transect was coincident with the medium point of each previous 100 m transect. Measurements were acquired at positions 1 m apart along the 20 m transects (Fig. 1c), overlapping the medium section of the 100 m transects."

**Line 107: how was the height assured?**

The height of measurement was assured by using a cart built specifically to carry the EM38 in the four positions - horizontally and vertically both at 0.15 m and 0.40 m. We revised the text to make it more clear.

**Old version (line 107)**

" $EC_a$  was collected at two heights from the soil surface (0.15 and 0.4 m) in the horizontal and vertical dipole orientations".

**New version**

" $EC_a$  was collected at two heights from the soil surface (0.15 and 0.4 m) in the horizontal and vertical dipole orientations, which was assured by placing the EM38 on a cart built specifically for this purpose. The cart has two shelves to accommodate the instrument, one at 0.15 m from the soil surface, and the other at 0.40 m from the soil surface."

Line 111: were the data interpolated to the same position at each transect? The data of each measurement date would differ. Specify the distance between the inversion positions.

Only the first two EMI surveys at each location were performed using a GPS to collect large amounts of data and to cover a bigger area to investigate spatial variations of  $EC_a$  at the beginning of the experiment. For these specific surveys, all readings at different heights and orientations were interpolated using an inverse distance to a power algorithm to the positions of the reference profile. We used a sampling rate of one measurement per second with speed walking of about 1 m/s resulting in sampling spacing of about 1 m. Subsequent EMI measurements were acquired **manually** at **1 m** spacing and along a small transect of 20 m for monitoring. No interpolation was required in this case as all readings at different heights and orientations were made at the same point.

Line 116: Details on the model are the least information that need to be included here. How many layers? Which depth? Fixed layer depths or is the layer thickness optimized as well?

The layer thickness in the inversion process is fixed and only the conductivity of each layer is optimized during the inversion process. We have revised the text to address the reviewer concern.

**Old version (line 115)**

"An Occam regularization (De Groot-Hedlin and Constable, 1990) based approach was used to invert the  $EC_a$  data. To run the algorithm, several parameters are selected, such as the type of inversion algorithm, the number of iterations, and the smoothing factor ( $\lambda$ ) that controls the roughness of the model. The optimal inversion parameters for the present conditions were obtained in previous studies for the study area (Farzamian et al., 2019)."

**New version**

"An Occam regularization (De Groot-Hedlin and Constable, 1990) based approach was used to invert the  $EC_a$  data. All  $EC_a$  data, collected at the four locations, were inverted by applying a five-layer earth initial model with electrical conductivity of 100 mS m-1 and a fixed layer thickness of 0.30 m. To run the algorithm, several parameters were selected, such as the type of

inversion algorithm, the number of iterations, and the smoothing factor ( $\lambda$ ) that controls the roughness of the model. The optimal inversion parameters for the present conditions were obtained in previous studies for the study area (Farzamian et al., 2019)."

**Line 119: "1.35m" - typo? 1.5 m is meant?**

1.35 m is correct. We revised the text to make it more clear.

**Old version (line 118)**

"At each sampling site, five soil samples were collected at 0.3 m increments to a depth of 1.35 m, (...)".

**New version**

"At each sampling site, five soil samples were collected at 0.3 m increments, from a depth of 0.15 m to 1.35 m, (...)".

**Line 129: what is the distance between the four locations?**

Location 1 to location 2: approximately 4 km; location 2 to location 3: approximately 5 km; location 3 to location 4: approximately 4.5 km. Location 1 to location 4: approximately 12.5 km.

**Line 130: "However"**

This word is needed here to express the fact that, although both calibrations are comparable, the regional one can be used expeditelly at any new location in the study area, which can't be done by the location-specific ones. No changes to the text were made relative to this comment.

**Line 133: $RMSE = 2.54 \text{ dS/m} \rightarrow error in slightly-saline range$**

The RMSE of 2.54 dS m-1 is good given the large range of ECe (0-37 dS m-1). We have revised the text and included the range of ECe in the study area.

**Old version (line 132)**

"The regional calibration was based on data collected during May and June 2017 and was validated using a leave-one-out-cross-validation method with good results ( $RMSE = 2.54 \text{ dS m}^{-1}$ )."

**New version**

"The regional calibration was based on data collected during May and June 2017 and was validated using a leave-one-out-cross-validation method with good results (RMSE =  $2.54 \text{ dS m}^{-1}$  in the 0–37 dS m-1 range)."

Figure 3: Please include on top of each figure:

left column: Transect i Theta (symbol)

right column: Transect i ECe

The figure shows  $\theta$  and ECe measured **in the sampling site** (pontual measurement) located at the middle of each transect, not  $\theta$  and ECe for the **entire transect**. Adding the transect number could be confusing for readers. Therefore we suggest revising the caption of the figure to make it more clear.

**Old version (caption of Fig.3)**

"Volumetric water content ( $\theta - m^3 m^{-3}$ ) and electrical conductivity of the soil saturation extract (ECe - 165 dS m-1), in the topsoil (0–0.3 m), subsurface (0.3–0.6 m), upper subsoil (0.6–0.9 m), intermediate subsoil (0.9–1.2 m), and lower subsoil (1.2–1.5 m), measured at the sampling site located in the middle of each transect, at locations 1 to 4, during the study period."

**New version**

"Volumetric water content ( $\theta - m^3 m^{-3}$ ) and electrical conductivity of the soil saturation extract (ECe - 165 dS m-1), in the topsoil (0–0.3 m), subsurface (0.3–0.6 m), upper subsoil (0.6–0.9 m), intermediate subsoil (0.9–1.2 m), and lower subsoil (1.2–1.5 m), measured in the sampling site located at the middle of each transect, at locations 1 to 4, during the study period. Each circled number refers to each location. Crosses refer to the dates when there were ECe measurements but no  $\sigma$  measurements, due to adverse field conditions."

**In the legend, please replace sigma by ECe**

The mention of  $\sigma$  in the plot's legend is correct. In fact, in those dates, we measured only ECe but we could not perform EM38 surveys due to the adverse field conditions. To make it more clear, we can add the following sentence in the figure caption: "Crosses refer to the dates when there were ECe measurements but no  $\sigma$  measurements, due to adverse field conditions."

**Old version (caption of Fig.3)**

"Volumetric water content ( $\theta - m^3 m^{-3}$ ) and electrical conductivity of the soil saturation extract (ECe - 165 dS m-1), in the topsoil (0–0.3 m), subsurface (0.3–0.6 m), upper subsoil (0.6–0.9 m), intermediate subsoil (0.9–1.2 m), and lower subsoil (1.2–1.5 m), measured at the sampling site located in the middle of each transect, at locations 1 to 4, during the study period."

**New version**

"Volumetric water content ( $\theta - m^3 m^{-3}$ ) and electrical conductivity of the soil saturation extract (ECe – 165 dS m-1), in the topsoil (0–0.3 m), subsurface (0.3–0.6 m), upper subsoil (0.6–0.9 m), intermediate subsoil (0.9–1.2 m), and lower subsoil (1.2–1.5 m), measured in the sampling site located at the middle of each transect, at locations 1 to 4, during the study period. Each circled number refers to each location. Crosses refer to the dates when there were ECe measurements but no  $\sigma$  measurements, due to adverse field conditions."

Line 168: The prediction results are totally unexpected here. Please show the inversion results first together with a description and profound discussion.

Thank you for the suggestion. Those results will be included as a new section, that will appear before the conversion of  $\sigma$  to ECe through the regional calibration. This section will be new section 4.2. Old section 4.2 will now be section 4.3, and old section 4.3 will be section 4.4. There will be a new Fig. 4. Old Fig. 4 will become Fig. 5, and old Fig. 5 will become Fig. 6.

**New version**

**"4.2. Time-lapse EMCIs**

Figure 4 shows the obtained EMCIs at locations 1 to 4 for each date of the EMI surveys. Globally,  $\sigma$  ranges from 19.44 mS m-1 to 1431.57 mS m-1 with the lowest values at location 1 and the highest at location 4. A general increasing trend of  $\sigma$  is quite evident from the north to the south, accompanying the previously known soil salinity gradient. In addition,  $\sigma$  increases with depth at locations 2, 3 and 4. At location 1,  $\sigma$  ranges spatiotemporally from 19.44 mS m-1 to 128.08 mS m-1. At location 2,  $\sigma$  ranges from 28.02 mS m-1 to 469.39 mS m-1 with highest values at depth. A similar pattern of  $\sigma$  is evident at locations 3 and 4. However, a greater range of  $\sigma$  is seen at location 3 with values from 36.23 mS m-1 to 706.32 mS m-1. Location 4 exhibits the largest variations of  $\sigma$ , ranging from 48.57 mS m-1 to 1431.57 mS m-1."

---

## Author Comment (AC2) · 17 Apr 2020

**"Monitoring soil salinity using time-lapse electromagnetic conductivity imaging" - authors' responses to suggestions and comments made in Public Discussion**

Dear editor and referees,

We sincerely appreciate and thank your constructive comments on our manuscript. We have revised the manuscript according to your suggestions and comments. We hope that the revised version of the manuscript properly addresses your concerns.

Our answers are placed in blue below each of the referees' interactive comments, and also below each of Referee #2's comments and corrections, provided in the supplementary annotated .pdf. The changes to the manuscript, arising from the referee's comments, are written in grey. We also added a section with some additional changes.

Sincerely,

Mohammad Farzamian on behalf of all authors

**Authors responses to referees' comments**

**Anonymous Referee #1**

https://doi.org/10.5194/soil-2019-99-RC1, 2020

In this paper, the authors explored the use of time-lapse EMI surveys and a 1-D inversion algorithm to predict soil salinity (ECe) at different depths at four sites in Lezíria de Vila Franca. A pre-established calibration equation was used to convert sigma to ECe. The model performance was good given the large R2, and Lin's concordance. They conclude that the method has the potential to be used for the rapid monitoring of soil salinity dynamics for agricultural management.

In general, the study is well-written and provides a nice example of monitoring soil salinity using non-invasive EMI surveys. I have a few comments about the manuscript.

We would like to thank Anonymous Referee #1 for evaluating our manuscript. We highly appreciate the overall positive comments.

1. The authors need to elaborate more on the pre-calibration of the EM38 meter. How are the variations of soil temperature (affecting soil ECa) be corrected?

The calibration of EM38 was done using the routine procedure (inphase nulling and instrument zero) recommended by the company at the beginning of measurements at each field.

We have also performed electrical resistivity tomography (ERT) surveys at the four locations and along the same EMI transects right after EMI surveys (2-3 times at each location), in order to evaluate by comparison the precision of the EMI data and models, as well as to explore the potential drift of the EM38 sensor. The ERT technique has usually a better precision in measuring soil electrical conductivity, but it is more effort- and time-consuming in the field, which limits its application for large-area investigations. The obtained ERT models agree with the EMCIs shown in this paper, which gives confidence to the EMCIs models. However, the results of this comparison are not in the scope of this paper.

Unfortunately, borehole temperature data were not available at the study sites (which are private farms) to correct the impact of soil temperature changes over the EM38 models. We agree that the lack of temperature correction will add to the uncertainty of temporal $EC_e$ changes assessment, although the impact of temperature changes over soil electrical conductivity is expected to be relatively small compared to soil salinity. We revised the text including the introduction, results and conclusion to address this issue.

**Old version (line 244)**

"Furthermore, the influence of static soil properties (i.e., that do not vary in time), such as clay content, could be tackled with the use of maps of the variation of soil salinity between two consecutive dates, which allows removing the static effect in the EMCIs."

**New version**

"In addition, the influence of static soil properties (i.e., that do not vary in time), such as clay content, could be tackled with the use of cross sections of the variation of soil salinity between two consecutive dates, which allows removing the static effect from the EMCIs. Future attempts can be made on finding ways to quantitatively account for the impact of dynamic soil properties (i.e. $\theta$ and temperature) on the EMCIs to optimize temporal soil salinity assessment."

**Old version (line 45)**

"EMI measures the apparent electrical conductivity of the soil ($EC_a$, dS m$^{-1}$), which is primarily a function of soil salinity, soil texture, water content, and cation exchange capacity; however, in a saline soil, soil salinity is generally the dominant factor responsible for the spatiotemporal variability of soil $EC_a$."

**New version**

"EMI measures the apparent electrical conductivity of the soil ($EC_a$, dS m$^{-1}$), which is a function of soil properties such as salinity, texture, cation exchange capacity, water content and temperature. However, in a saline soil, soil salinity is generally the dominant factor responsible for the spatiotemporal variability of soil $EC_a$."

**Old version (line 183)**

"Furthermore, this test set is composed of measurements collected over a wider period of time (18 months). During this period, soil properties, which are also known to influence $\sigma$, such as $\theta$, change (as shown in Fig. 3), which introduces larger variability in the measurements."

**New version**

"Furthermore, this test set is composed of measurements collected over a wider period of time (18 months). During this period, soil properties, which are also known to influence $\sigma$, such as temperature and $\theta$, change , which introduces larger variability in the measurements."

2. Please discuss more on the pre-determined linear calibration equation. Since the soil water content is measured, can you include this in the model to improve the model performance? Can the variations in soil water content or soil texture explain the small misfit between the predicted ECe and measured ECe?

The water content could be included along with σ in the model for prediction of EC$_e$. As these variables are linearly correlated, this could be achieved through a principal component regression (PCR). But in this paper our aim was to test the validity of the regional calibration over time, which includes wide variations of water content and other conditions, because the strength of the regional calibration is that it is very practical for monitoring salinity in this large area. If the water content would be included as a predictor in the model, it would have to be measured along with σ in the field, which would increase the complexity of the monitoring process.

In previous studies we also found that, in this study area, the water content is correlated with σ, but that properties related to salinity (EC$_e$, SAR and ESP) have a larger influence on σ. We propose to add the following text for further clarifying this aim and its limitations. We also strengthen the importance of validating the previous approach with an independent test set, adding a new paragraph.

**Old version (line 59)**

"(...) and sodium adsorption ratio."

**New version**

"(...) and sodium adsorption ratio. In this later study, the authors performed a principal component analysis of the soil properties in the study area, and found that the water content was correlated with sigma, but with a relatively lower influence when compared to the properties related to salinity (EC$_e$, SAR and ESP).

Because the inversion of EC$_a$ is relatively recent since the use of EMI for soil characterization, the lack of validation using an independent data set still limits the use of this new methodology (Corwin and Scudiero, 2019), making it therefore important to further test its accuracy in salinity monitoring."

The reference will also be added to the references list.

"Corwin, D.L. and Scudiero, E.: Chapter One - Review of soil salinity assessment for agriculture across multiple scales using proximal and/or remote sensors. Adv Agron, 158, 1–130, https://doi.org/10.1016/bs.agron.2019.07.001, 2019."

The soil electrical conductivity is a complex function of soil properties, including soil texture, water content and salinity. Therefore, water content and soil texture could impact the misfit between the predicted EC$_e$ and measured EC$_e$.

3. If possible, is it possible to include soil water content in the calibration model and develop a pedo-transfer physical model coupled with a novel inversion algorithm to simultaneously retrieve soil water and ECe values from time-lapse EMI surveys of ECa values in the future?

It is mathematically possible to develop such an inversion algorithm, but it goes quite beyond the scope of the present paper. It is worth mentioning that one can not estimate both water content and $EC_e$ from soil electrical conductivity alone as both parameters influence soil electrical conductivity. On the other hand, including water content in the $EC_e$ estimation and possible inversion algorithm requires water content measurements along with EMI measurements in the field. Please see also our answer to the previous question.

**Anonymous Referee #2**

https://doi.org/10.5194/soil-2019-99, 2020

Dear Maria Catarina Paz et al., the study entitled 'Monitoring soil salinity using time-lapse electromagnetic conductivity imaging' provides an overview of salinity assessment using EMI data and inversions together with ground-truth data in an agricultural area close to Lisboa. In general, the manuscript is very well written and the narrative is nicely presented.

With this review, I enclosed some comments in the annotated .pdf ('soil-2019-99supplement.pdf') that is provided with the review process. Please consider the comments in that pdf for further improvements. Including my aspects together with the suggestions of the previous reviewer, I'd recommend publishing the manuscript in SOIL.

Kind regards, Referee 2

We would like to thank Anonymous Referee #2 for evaluating our manuscript. We highly appreciate the overall positive comments.

Supplementary .pdf

Line 24: ""

OK. We have revised the text accordingly.

**Old version (line 24)**

"The results showed that ECe was satisfactorily predicted, (…)".

**New version:**

"The results showed that ECe was predicted, (…)".

Line 47: salinity is dominant as long as soil is moist

We agree and have revised the text accordingly.

**Old version (line 47)**

"(…) however, in saline soil, soil salinity is generally the dominant factor responsible for the spatiotemporal variability of soil $EC_a$."

**New version**

"(…) however, in saline soil, soil salinity is generally the dominant factor responsible for the spatiotemporal variability of soil $EC_a$ when soil is moist."

Line 54: please include here some more studies that use EMI and EMI inversions. e.g., Corwin&Scudiero (Advances in Agronomy), von Hebel et al. 2019 (sensors), Kaufmann et al. 2020 (Soil Use and management), Wang et al. 2019 (JGR), Guillemoteau et al. 2019 (GJI)

We have now included more references.

**Old version (line 47)**

"EMI surveys have been successfully used in conjunction with soil sampling to assess soil salinity through location-specific calibration between measured $EC_a$ and soil salinity (e.g. Triantafilis et al., 2000; 2001; Corwin and Lesch, 2005; Bouksila et al., 2012)."

**New version**

"EMI surveys have been successfully used in conjunction with soil sampling to assess soil salinity through location-specific calibration between measured ECa and soil salinity (e.g. Triantafilis et al., 2000; 2001; Corwin and Lesch, 2005; Bouksila et al., 2012; Corwin and Scudiero, 2019; Guillemoteau et al., 2019; von Hebel et al. 2019; Wang et al., 2019; Kaufmann et al. 2020)."

Line 62: Please rephrase the sentence that no study maps salinity. Actually, this is why the EM38 was invented back in the 1980-ties. Numerous EMI studies map salinity. Also with inversion. E.g., by Triantafillis group.

We agree that many studies inverted EMI data for soil salinity mapping. However, the sentence relates to the inversion of **time-lapse** data for **monitoring** soil salinity in time which has not been reported yet to the best of our knowledge. No changes to the text were made relative to this comment.

Line 103: Although the EM38 is well known, please include the basic information like coil distance, coil orientation, depth range of investigation. This also help "new readers" guiding through the inversion results.

Agreed. We have revised the text accordingly.

**Old version (line 103)**

"EMI data was acquired using the EM38 instrument (Geonics Ltd, Mississauga, Canada) on five dates at locations 1 and 4, and on six dates at locations 2 and 3, during the period of May 2017 to October 2018."

**New version**

"EMI data was acquired using the EM38 instrument (Geonics Ltd, Mississauga, Canada). Technology of this instrument is based on two coils, one transmitting the electromagnetic signal, and the other receiving it, distanced 1 m apart from each other inside the instrument case. The position of these coils can be controlled by placing the instrument in a vertical position relative to the soil surface - horizontal dipole mode (the coils stand in the horizontal position), which provides a maximum depth of investigation of 1.5 m - or in a horizontal position relative to the soil surface - vertical dipole mode (the coils stand in the vertical position), which provides a maximum depth of investigation of 0.75 m. EM38 surveys were done on five dates at locations 1 and 4, and on six dates at locations 2 and 3, during the period of May 2017 to October 2018."

Line 105: accuracy of GPS?

GPS has 5 m position accuracy. We can add the accuracy of the GPS to the sentence.

**Old version (line 104)**

"Measurements on the two first dates were continuously acquired along 100 m transects using a GPS (Rikaline 6010) for registration of the position."

**New version**

"Measurements on the two first dates were continuously acquired at each location, along a 100 m transect, using a GPS (Rikaline 6010, with 5 m position accuracy) for registration of the position."

Line 106: unclear description of measurement procedure along/at the transects

We have revised the text accordingly.

**Old version (line 105)**

"Subsequent EMI measurements were acquired at positions 1 m apart along 20 m transects (Fig. 1c), overlapping the medium section of the 100 m transects."

**New version**

"Subsequent EMI measurements were acquired at each location, along a 20 m transect. The middle point of each 20 m transect was coincident with the medium point of each previous 100 m transect. Measurements were acquired at positions 1 m apart along the 20 m transects (Fig. 1c), overlapping the medium section of the 100 m transects."

Line 107: how was the height assured?

The height of measurement was assured by using a cart built specifically to carry the EM38 in the four positions - horizontally and vertically both at 0.15 m and 0.40 m. We revised the text to make it more clear.

"$EC_a$ was collected at two heights from the soil surface (0.15 and 0.4 m) in the horizontal and vertical dipole orientations".

**New version**

 "$EC_a$ was collected at two heights from the soil surface (0.15 and 0.4 m) in the horizontal and vertical dipole orientations, which was assured by placing the EM38 on a cart built specifically for this purpose. The cart has two shelves to accommodate the instrument, one at 0.15 m from the soil surface, and the other at 0.40 m from the soil surface."

Line 111: were the data interpolated to the same position at each transect? The data of each measurement date would differ. Specify the distance between the inversion positions.

Only the first two EMI surveys at each location were performed using a GPS to collect large amounts of data and to cover a bigger area to investigate spatial variations of $EC_a$ at the beginning of the experiment. For these specific surveys, all readings at different heights and orientations were interpolated using an inverse distance to a power algorithm to the positions of the reference profile. We used a sampling rate of one measurement per second with speed walking of about 1 m/s resulting in sampling spacing of about 1 m. Subsequent EMI measurements were acquired **manually** at **1 m** spacing and along a small transect of 20 m for monitoring. No interpolation was required in this case as all readings at different heights and orientations were made at the same point.

Line 116: Details on the model are the least information that need to be included here. How many layers? Which depth? Fixed layer depths or is the layer thickness optimized as well?

The layer thickness in the inversion process is fixed and only the conductivity of each layer is optimized during the inversion process. We have revised the text to address the reviewer concern.

**Old version (line 115)**

"An Occam regularization (De Groot-Hedlin and Constable, 1990) based approach was used to invert the $EC_a$ data. To run the algorithm, several parameters are selected, such as the type of inversion algorithm, the number of iterations, and the smoothing factor ($\lambda$) that controls the roughness of the model. The optimal inversion parameters for the present conditions were obtained in previous studies for the study area (Farzamian et al., 2019)."

**New version**

"An Occam regularization (De Groot-Hedlin and Constable, 1990) based approach was used to invert the $EC_a$ data. All $EC_a$ data, collected at the four locations, were inverted by applying a five-layer earth initial model with electrical conductivity of 100 mS m$^{-1}$ and a fixed layer thickness of 0.30 m. To run the algorithm, several parameters were selected, such as the type of

inversion algorithm, the number of iterations, and the smoothing factor ($\lambda$) that controls the roughness of the model. The optimal inversion parameters for the present conditions were obtained in previous studies for the study area (Farzamian et al., 2019)."

Line 119: "1.35m" - typo? 1.5 m is meant?

1.35 m is correct. We revised the text to make it more clear.

**Old version (line 118)**

"At each sampling site, five soil samples were collected at 0.3 m increments to a depth of 1.35 m, (...)".

**New version**

"At each sampling site, five soil samples were collected at 0.3 m increments, from a depth of 0.15 m to 1.35 m, (...)".

Line 129: what is the distance between the four locations?

Location 1 to location 2: approximately 4 km; location 2 to location 3: approximately 5 km; location 3 to location 4: approximately 4.5 km. Location 1 to location 4: approximately 12.5 km.

Line 130: ""

This word is needed here to express the fact that, although both calibrations are comparable, the regional one can be used expeditelly at any new location in the study area, which can't be done by the location-specific ones. No changes to the text were made relative to this comment.

Line 133: RMSE = 2.54 dS/m -> error in slightly-saline range

The RMSE of 2.54 dS m$^{-1}$ is good given the large range of EC$_e$ (0-37 dS m$^{-1}$). We have revised the text and included the range of EC$_e$ in the study area.

**Old version (line 132)**

"The regional calibration was based on data collected during May and June 2017 and was validated using a leave-one-out-cross-validation method with good results (RMSE = 2.54 dS m$^{-1}$)."

**New version**

"The regional calibration was based on data collected during May and June 2017 and was validated using a leave-one-out-cross-validation method with good results (RMSE = 2.54 dS m$^{-1}$ in the 0–37 dS m$^{-1}$ range)."

Figure 3: Please include on top of each figure:

left column: Transect i Theta (symbol)

right column: Transect i ECe

The figure shows $\theta$ and $EC_e$ measured **in the sampling site** (pontual measurement) located at the middle of each transect, not $\theta$ and $EC_e$ for the **entire transect**. Adding the transect number could be confusing for readers. Therefore we suggest revising the caption of the figure to make it more clear.

**Old version (caption of Fig.3)**

"Volumetric water content ($\theta$ – m$^3$ m$^{-3}$) and electrical conductivity of the soil saturation extract ($EC_e$ – 165 dS m$^{-1}$), in the topsoil (0–0.3 m), subsurface (0.3–0.6 m), upper subsoil (0.6–0.9 m), intermediate subsoil (0.9–1.2 m), and lower subsoil (1.2–1.5 m), measured at the sampling site located in the middle of each transect, at locations 1 to 4, during the study period."

**New version**

"Volumetric water content ($\theta$ – m$^3$ m$^{-3}$) and electrical conductivity of the soil saturation extract ($EC_e$ – 165 dS m$^{-1}$), in the topsoil (0–0.3 m), subsurface (0.3–0.6 m), upper subsoil (0.6–0.9 m), intermediate subsoil (0.9–1.2 m), and lower subsoil (1.2–1.5 m), measured in the sampling site located at the middle of each transect, at locations 1 to 4, during the study period. Each circled number refers to each location. Crosses refer to the dates when there were $EC_e$ measurements but no $\sigma$ measurements, due to adverse field conditions."

In the legend, please replace sigma by ECe

The mention of $\sigma$ in the plot's legend is correct. In fact, in those dates, we measured only $EC_e$ but we could not perform EM38 surveys due to the adverse field conditions. To make it more clear, we can add the following sentence in the figure caption: "Crosses refer to the dates when there were $EC_e$ measurements but no $\sigma$ measurements, due to adverse field conditions."

**Old version (caption of Fig.3)**

"Volumetric water content ($\theta$ – m$^3$ m$^{-3}$) and electrical conductivity of the soil saturation extract ($EC_e$ – 165 dS m$^{-1}$), in the topsoil (0–0.3 m), subsurface (0.3–0.6 m), upper subsoil (0.6–0.9 m), intermediate subsoil (0.9–1.2 m), and lower subsoil (1.2–1.5 m), measured at the sampling site located in the middle of each transect, at locations 1 to 4, during the study period."

**New version**

"Volumetric water content ($\theta$ – $m^3$ $m^{-3}$) and electrical conductivity of the soil saturation extract ($EC_e$ – 165 dS $m^{-1}$), in the topsoil (0–0.3 m), subsurface (0.3–0.6 m), upper subsoil (0.6–0.9 m), intermediate subsoil (0.9–1.2 m), and lower subsoil (1.2–1.5 m), measured in the sampling site located at the middle of each transect, at locations 1 to 4, during the study period. Each circled number refers to each location. Crosses refer to the dates when there were $EC_e$ measurements but no $\sigma$ measurements, due to adverse field conditions."

Line 168: The prediction results are totally unexpected here. Please show the inversion results first together with a description and profound discussion.

Thank you for the suggestion. Those results will be included as a new section, that will appear before the conversion of $\sigma$ to $EC_e$ through the regional calibration. This section will be new section 4.2. Old section 4.2 will now be section 4.3, and old section 4.3 will be section 4.4. There will be a new Fig. 4. Old Fig. 4 will become Fig. 5, and old Fig. 5 will become Fig. 6.

**New version**

"**4.2. Time-lapse EMCIs**

Figure 4 shows the obtained EMCIs at locations 1 to 4 for each date of the EMI surveys. Globally, $\sigma$ ranges from 19.44 mS $m^{-1}$ to 1431.57 mS $m^{-1}$ with the lowest values at location 1 and the highest at location 4. A general increasing trend of $\sigma$ is quite evident from the north to the south, accompanying the previously known soil salinity gradient. In addition, $\sigma$ increases with depth at locations 2, 3 and 4. At location 1, $\sigma$ ranges spatiotemporally from 19.44 mS $m^{-1}$ to 128.08 mS $m^{-1}$. At location 2, $\sigma$ ranges from 28.02 mS $m^{-1}$ to 469.39 mS $m^{-1}$ with highest values at depth. A similar pattern of $\sigma$ is evident at locations 3 and 4. However, a greater range of $\sigma$ is seen at location 3 with values from 36.23 mS $m^{-1}$ to 706.32 mS $m^{-1}$. Location 4 exhibits the largest variations of $\sigma$, ranging from 48.57 mS $m^{-1}$ to 1431.57 mS $m^{-1}$."

[Figure]

**Figure 4: Time-lapse electromagnetic conductivity images (EMCIs) for locations 1 to 4.**

Line 186: This high ECe (52.35 dS/m) is only at the location 4. Here, a more realistic conclusion for the larger study site that includes transect 1-3 with much lower ECe is lacking. Please add.

We thank you for your suggestion. The regional calibration has been developed to be used at any new location in the entire Lezíria region. It's true that the prediction ability is not so satisfactory in the northern location, but still it works as an expedit method for soil salinity prediction in Lezíria. For better prediction abilities, location-specific calibrations can be developed using data from each location. However, this limits the application of the calibration to each specific location and this is not the objective of this paper. We have revised the text to address the reviewer concern and added a new paragraph at line 181:

"The prediction ability of the regional calibration can be less satisfactory when analysing the performance for specific locations within the region, where the ranges of $EC_e$ are smaller, particularly in the less saline northern locations. However, the calibration has been developed using data from the entire region, and it has been independently validated with data from that same region, with the purpose of using it for expedit soil salinity prediction in the entire study area. For optimal prediction ability at each specific location, location-specific calibrations can be developed, as done for instance in Farzamian et al. (2019)."

Line 188: With the last sentence, it seems you can perform that calibration, since you have the data. Otherwise please make clear why this is not the case.

We proposed in that sentence a continuation of the monitoring to improve the calibration and we do not have more data at this stage. We revised the text to avoid the confusion for the readers.

**Old version (line 186)**

"The regional calibration could be further developed by including measurements taken over a longer period of time in the calibration process, in order to include a wider range of variation of soil properties."

**New version**

"The regional calibration could be further refined by repeating measurements over a longer period of time in the calibration process, in order to include a wider range of variation of soil properties. In addition, monitoring at new locations with different $EC_e$ ranges could also be included to improve the prediction ability of the regional calibration."

Line 196: "maps" - The figures present a profile and not a map. Please revise and use a word like profile, transect, or cross section throughout the manuscript.

OK, we can use the term **cross section** instead of the word **map.** The entire manuscript will be revised accordingly.

In particular, we revised the title of section 4.3.

**Old version (line 195)**

"**4.3 Spatiotemporal mapping of soil salinity from time-lapse EMCI**"

**New version**

"**4.4 Spatiotemporal imaging of soil salinity from time-lapse EMCI**"

Figure 5: please also show the legend beneath cross sections of transect 1

We used a single scale legend for all four locations for an easier comparison for readers.  We revised the figure and included the scale in the middle of the figure.

**Old version (Fig. 5)**

[Figure]

**New version (now Fig. 6)**

[Figure]

ECₑ (dS m⁻¹)

Line 213: At transect 2 for October 2017, please describe the differences in the topsoil between measured (moderately saline) and modelled (non-saline) data.

The underestimation of modelled $EC_e$ is due to the measured soil salinity of 2.03 dS m⁻¹, classified as slightly saline but still very close to the non-saline classification. No changes to the text were made relative to this comment.

Line 219: For transect 3, please describe why the cross sections show 2 saline layers in Jan 2018 and July 2018. Also due to fertigation as in transect 2?

We think that the pattern seen at location 3 in July 2018, as well as the similar one at location 2 on the same date, are due to the fertigation practices for the maize cultivation. We have revised the text to address this issue. This is not the case for the cross section seen in January 2018, when there is a non-irrigated winter crop with no fertigation practices. The rise of groundwater (0.9 m) in this period and rainfall events before the survey are expected to be the main reasons for the January 2018 pattern.

**Old version (line 220)**

"The groundwater level is above 1.5 m in January 2018, although the salinity of the deeper soil layers (>0.9 m) decreases compared to May and October 2017, which could be due to washing of the profile by rainfall."

**New version**

"The groundwater level is above 1.5 m in January 2018, although the salinity of the deeper soil layers (>0.9 m) decreases compared to May and October 2017, which could be due to washing of the profile by rainfall. The increase of soil salinity in upper soil layers in July 2018, similarly to location 2 on the same date, can be attributed to fertigation practices for the maize cultivation."

Line 229: please make two sentences here.

OK. The text has been revised accordingly.

**Old version (line 229)**

"Location 1 tends to have non-saline characteristics, which can be attributed to good quality irrigation water and to the fact that this location is far from the estuary, making it less prone to the presence of saline groundwater."

**New version**

"Location 1 tends to have non-saline characteristics, which can be attributed to good quality irrigation water. In addition, this location is far from the estuary, making it less prone to the presence of saline groundwater."

Line 239: please 2 sentences.

OK. The text has been revised accordingly.

**Old version (line 238)**

"This validation resulted in lower prediction ability than that previously resulting from cross-validation, not only because the test set was independent, but also because it was collected over a wider period of time, during which the variation of soil properties is larger."

**New version**

"This validation resulted in lower prediction ability than that previously resulting from cross-validation. This is because the test set was independent, and also because it was collected over a wider period of time, with a larger variation of soil properties."

Line 241: this conclusion might change. see above

Answered above.

**Additional changes**

We would also like to ask for changing the following group of sentences.

**Old version (line 139)**

"The RMSE is the square root of the mean of the squared differences between the measured and predicted $EC_e$, indicating how concentrated the data is around the linear regression. In this study we used two degrees of freedom for a more robust calculation of RMSE. The coefficient of determination (R2) indicates how well the predicted $EC_e$ approximate the measured $EC_e$. When this is 1, it means the predictions coincide with the measurements. Lin's CCC measures the agreement between the measured and predicted ECe evaluating how close the linear regression is to the 1:1 relationship and ranges from −1 to 1, with perfect agreement at 1 (Lin, 1989)."

**New version**

"The RMSE is the square root of the mean of the squared differences between the measured and predicted $EC_e$. In this study we used two degrees of freedom for a more robust calculation of RMSE. ME is the mean of all differences between the measured and predicted $EC_e$ and evaluates whether the linear regression consistently over- and underestimates the predicted $EC_e$. Therefore, the prediction is more precise and less biased when the RMSE and the ME are closer to zero. The coefficient of determination ($R^2$) indicates the proportion of the variance that is predicted by the model. When this is 1, it means the model totally explains the variation. Lin's CCC measures the agreement between the measured and predicted $EC_e$ evaluating their deviation from the 1:1 relationship and ranges from −1 to 1, with perfect agreement at 1 (Lin, 1989)."

Also, one of the references (previously referred to as Paz at al., 2019a) which was in press, has been updated. Therefore, all citations in the text referring to Paz et al. (2019a) will be changed to Paz et al. (2020), and all citations in the text referring to Paz et al. (2019b) will be changed to Paz et al. (2019).

**Old version (line 316)**

"Paz, A., Castanheira., N., Farzamian, M., Paz, M.C., Gonçalves, M., Monteiro Santos, F., and Triantafilis, J.: Prediction of soil salinity and sodicity using electromagnetic conductivity imaging. Geoderma, https://doi.org/10.1016/j.geoderma.2019.114086, *in press*, 2019a."

"Paz, M.C., Farzamian, M., Monteiro Santos, F., Gonçalves, M.C., Paz, A.M., Castanheira., N.L., and Triantafilis, J.: Potential to map soil salinity using inversion modelling of EM38 sensor data. First Break, 37(6), 35–39, doi:10.3997/1365-2397.2019019, 2019b."

**New version**

"Paz, A., Castanheira., N., Farzamian, M., Paz, M.C., Gonçalves, M., Monteiro Santos, F., and Triantafilis, J.: Prediction of soil salinity and sodicity using electromagnetic conductivity imaging. Geoderma, 361, 114086, https://doi.org/10.1016/j.geoderma.2019.114086, 2020."

"Paz, M.C., Farzamian, M., Monteiro Santos, F., Gonçalves, M.C., Paz, A.M., Castanheira., N.L., and Triantafilis, J.: Potential to map soil salinity using inversion modelling of EM38 sensor data. First Break, 37(6), 35–39, doi:10.3997/1365-2397.2019019, 2019."

Finally, we detected a lapse in one of the references. The reference has been revised.

**Old version (line 279)**

Corwin, D.L. and Lesch, S.M.: Characterizing soil spatial variability with apparent soil electrical conductivity: I. Survey protocols. Comp. Elec. Agri. Appl. Apparent Soil Elec. Conductivity Precis. Agri., 46, 103–133, https://doi.org/10.1016/j.compag.2004.11.002, 2005.

**New version**

Corwin, D.L. and Lesch, S.M.: Characterizing soil spatial variability with apparent soil electrical conductivity: I. Survey protocols. Comp. Elec. Agri., 46, 103–133, https://doi.org/10.1016/j.compag.2004.11.002, 2005.

---

## Author Response (AR1)

**"Monitoring soil salinity using time-lapse electromagnetic conductivity imaging" - authors' responses to suggestions and comments made by Topical Editor Jan Vanderborght**

Dear topical editor Jan Vanderborght,

We sincerely appreciate and thank your constructive comments on our manuscript. We have revised the manuscript according to your suggestions and comments. We hope that the revised version of the manuscript properly addresses your concerns.

Our answers are placed in blue below each of your comments. The changes to the manuscript, arising from your comments, are written in grey.

Sincerely,
Mohammad Farzamian on behalf of all authors

Authors responses to topical editor Jan Vanderborght's comments

Comments to the Author:

Dear,

I agree with your replies to the comments of the reviewers.

However, I have two main additional comments.

We thank you very much for pointing out these two very relevant and constructive questions.

The first is on the statistical evaluation of agreement between the measured and EMI derived ECe's. This was to my opinion not clearly described and I propose including the equations that were used to calculate the RMSEs, R2 and other indices. The reason for including these is that there is an ambiguity about what you use as predicted value when you calculate these indices. The ECe that you obtained from the EMI estimated sigma and a regression equation, or the prediction of the measurement from a regression between the EMI derived ECe and the measured ECe?

We have added a table with the equations used for calculating the statistical indicators, and also revised section 2.4 as follows:

**Old version (line 135–in section 2.4)**

"In the present study, the regional calibration was used to predict $EC_e$ from time-lapse EMCI. The predicted $EC_e$ and $EC_e$measured from soil samples, collected from July 2017 to October 2018, were used to validate the regional calibration as anindependent test set.Its prediction ability was evaluated by calculating the root mean square error (RMSE), the coefficient ofdetermination ($R^2$) between the measured and predicted $EC_e$, the Lin's concordance correlation coefficient (CCC), and the mean error (ME).The RMSE is the square root of the mean of the squared differences between the measured and predicted $EC_e$, indicating how concentrated the data is around the linear regression. In this study we used two degrees of freedom for a more robust calculation of RMSE. The coefficient of determination ($R^2$) indicates how well the predicted $EC_e$ approximate the measured $EC_e$. When this is 1, it means the predictions coincide with the measurements. Lin's CCC measures the agreement between the measured and predicted $EC_e$ evaluating how close the linear regression is to the 1:1 relationship and ranges from −1 to 1, with perfect agreement at 1 (Lin, 1989). ME is the mean of all differences between the measured and predicted $EC_e$ and evaluates whether the linear regression consistently over- and underestimates the predicted ECe. Therefore, the prediction is more precise and less biased when the RMSE and the ME are closer to zero."

**New version**

"In the present study, the regional calibration was used to predict$EC_e$ from time-lapse EMCI ($pEC_e$). The predicted $EC_e$ and $EC_e$measured from soil samples ($mEC_e$), collected at the same time as the EMI surveys, were used as an independent data set for the validation of the regional calibration. The validation was performed by calculating the root mean square error (RMSE), the coefficient of determination ($R^2$) between the measured and predicted $EC_e$, the Lin's concordance correlation coefficient (CCC), and the mean error (ME). Description of these statistical indicators and the equations used to calculate them are shown in Table 1. Calculations were done using global data, and also using data discriminated by date of measurement (in this case we considered dates when measurements were done at the four locations – January, June and October 2018), depth of measurement and location."

**Table 1 – Description and equations of the statistical indicators used to evaluate the prediction ability of the regional calibration in this work.**

| Statistics | Equation[1] | Description |
|---|---|---|
| Root mean square error (RMSE) | $$RMSE = \sqrt{\frac{\sum_{i=1}^{n}(mEC_{e_i} - pEC_{e_i})^2}{n-2}}$$ | Evaluates matching between measured and predicted data. When it is zero, it indicates perfect matching between measured and predicted data. |
| Mean error (ME) | $$ME = \frac{\sum_{i=1}^{n}(mEC_{e_i} - pEC_{e_i})}{n}$$ | Evaluates whether the predicted data are over- or underestimated. A negative value means overestimation, a positive value means underestimation. |
| Lin's concordance correlation coefficient (Lin's CCC) | $$Lin's\ CCC = \frac{2s_{mEC_e - pEC_e}}{s_{mEC_e}^2 + s_{pEC_e}^2 + (\overline{mEC_{e_i}} - \overline{pEC_{e_i}})^2}$$ | Evaluates agreement between measured and predicted data. Ranges from -1 to 1. When it is 1, it indicates perfect agreement between measured and predicted data (Lin, 1989). |
| Coefficient of determination (R²) | $$R^2 = \left( \frac{\sum_{i=1}^{n}(mEC_{e_i} - \overline{mEC_{e_i}})(pEC_{e_i} - \overline{pEC_{e_i}})}{\sqrt{\sum_{i=1}^{n}(mEC_{e_i} - \overline{mEC_{e_i}})^2}\ \sqrt{\sum_{i=1}^{n}(pEC_{e_i} - \overline{pEC_{e_i}})^2}} \right)^2$$ | Indicates the degree of linearity between predicted and measured data. Ranges from 0 to 1. Above 0.5 is considered satisfactory. |

[1] n is the total number of data; $mEC_e$ is measured $EC_e$; $pEC_e$ is predicted $EC_e$; the upper bar represents the mean of the indicated data.

The second main (and more critical) comment is that you do not address the real issue of this paper, namely, can you use EMI to MONITOR ECe, i.e. to evaluate whether it changes over time. You are only discussing that you observe changes over time. But, are the changes you observe in the soil sample measurements consistent with the changes that you observe in the EMI measurements? You do not address this issue and I think this is crucial. A negative answer to this question will not lead to a rejection of your paper. But, you need to answer this question using appropriate statistical analyses. If this question is not answered appropriately, the work that you present is not meeting the objectives that are suggested in the title, the introduction and the conclusion and can therefore not be accepted.

Relatively to the second question, we added specific statistical analyses to investigate the prediction ability of the calibration in time and at different locations and depths to address the editor´s concern. The new analysis shows that the prediction ability of the regional calibration does not vary significantly over time. However, the changes in predicted $EC_e$ are not completely consistent with the changes in measured $EC_e$. This is partly due todifferent spatial and temporal variations of other soil properties at each location which influence spatio-temporal variations of σ differently and limit inferring soil salinity changes using time-lapse EMCI data and regional calibration, as pointed out by the editor.On the other hand, the regional calibration is more influenced by the spatial variability of $EC_e$ at location 4 which also limits a quantitative investigation of soil salinity changes from time-lapse EMI surveys at other locations.

The new statistical analyses added to the revised version of the manuscript suggests that the regional calibration approach still stands as an expeditious method to predict soil salinity from EMI in the study area (any new locations in the study area), since the prediction ability of the regional calibration has not been changed significantly. However, a location-specific calibration is required for a more precise assessment of soil salinity changes at each location.We also detected an outlier (in the lower subsoil at location 2), and took it out – this changed slightly the results presented for the global statistical indicators. We revised the text and Fig. 4 in this light and we propose a new title for the paper which may better represent the results of this study.

**Old version (Title)**

[revised manuscript text omitted]

Detailed comments:

Ln 15: You could maybe include here why the soil faces a risk of salinization in this area.

The soil faces the risk of salinization due to the influence of the ocean tides in groundwater in the southern part of the peninsula. Irrigation can also represent a salinization risk at the northern locations. Climate change is likely to aggravate these risks through the rise of sea water level (resulting in the rise of the saline groundwater and increased salinity of the irrigation water, which is collected from the river, upstream of the estuary), the increase in temperature and decrease in rainfall, which can lead to the accumulations of salts in the soil profile.

The proposed change is presented below.

**Old version (line 14 – in Abstract)**

"Lezíria Grande of Vila Franca de Xira, located in Portugal, is an important agricultural system where soil faces the risk of salinization, being thus prone to desertification and land abandonment."

**New version**

"Lezíria Grande of Vila Franca de Xira, located in Portugal, is an important agricultural system where soil faces the risk of salinization due to climate change, as the level and salinity of groundwater are likely to increase, as a result of the rise of the sea water level and consequently of the estuary. These changes can also affect the salinity of the irrigation water which is collected upstream of the estuary."

To avoid confusion, I propose to use the same units for ECa, sigma, and ECe.

In fact, there was a lapse in the units of $EC_a$, thank you for noticing it. It should be in mS m$^{-1}$. The manuscript will be corrected accordingly.

Relatively to $EC_e$ and σ, we maintained the same units, since the calibration equation that is a substantial part of this paper was parameterized for $EC_e$ in dS m$^{-1}$ and σ in mS m$^{-1}$.

Ln 23: 'This study aims to evaluate the potential of time-lapse EMCI and the regional calibration to predict the spatiotemporal variability of soil salinity in the study area.' After having read the article, I think that this statement is not sufficiently supported by the results that were presented and discussed. You indeed evaluated whether time lapse EMI measurements can explain the total variation in a dataset of measured ECe values at different locations and depths and at different times. But, the question whether EMI can be used to monitor changes in ECe over time was not addressed and to my opinion it should be. You should evaluate whether the changes in ECe that are observed over time correspond with changes in EMI estimated ECe over time. When the answer to this question is negative, this paper shows that the spatial variations in ECe in the studied area are much larger than the temporal variations so that EMI can still be used to map spatial variations but not the temporal variability due to irrigation and seasonal leaching. Evaluating the temporal variability of ECe within a year might require to have also information about variation of soil water content, groundwater table depth and soil temperature, which also vary over time and modulate the variations of ECe over time.

Please see our detailed answer to the previous question and corresponding revision in this regard.

Ln 27: 'revealed salinity fluctuations related to the input of salts and water either through irrigation, precipitation or groundwater level and salinity.' You best reformulate this sentence. 'Salinity fluctuation related to salinity' is a circular explanation.

It refers to the salinity of groundwater. We revised the sentence to clarify this issue:

**Old version (line 28)**

"(...) through irrigation, precipitation or groundwater level and salinity."

**New version**

"(...) through irrigation, precipitation or level and salinity of groundwater."

Ln 100: How was reference ET obtained?

ET is calculated by the Penman-Monteith method and is given by information collected in the meteorological station shown in Figure 1.

Ln 119: Can you give some details about the soil samples? Soil rings or bulk samples? Which volume, mass of soil?

Bulk density (g m$^{-3}$) was determined once for each soil layer from undisturbed soil samples (100 cm$^3$) oven-dried at 105ºC. $EC_e$ was measured in the extract collected with suction filters from disturbed soil samples, according to the method described by Richards (1954).

[Richards, L.A. (Ed.), 1954. Diagnosis and Improvement of Saline and Alkali Soils. Agricultural Handbook, USDA]

**Old version (line 122)**

"(...) after bulk density determination."

**New version**

"(...) after bulk density (g m$^{-3}$) determination from undisturbed 100 cm$^3$ soil samples."

**Old version (line 122)**

"EC$_e$ was measured with a conductivity meter (WTW 1C20-0211 inoLab) in the extract collected from the soil saturation paste."

**New version**

"EC$_e$ was measured with a conductivity meter (WTW 1C20-0211 inoLab) in the extract collected from the soil saturation paste obtained from 300 g of air-dry soil samples, according to the methods described by Richards (1954)."

**Old version (line 321)**

"        (...) 2397.2019019, 2019b.

Shanahan, P.W. (...)"

**New version**

"        (...) 2397.2019019, 2019b.

Richards, L.A. (Ed.), 1954. Diagnosis and Improvement of Saline and Alkali Soils. Agricultural Handbook, USDA.

Shanahan, P.W. (...)"

Ln 139: 'The RMSE is the square root of the mean of the squared differences between the measured and predicted ECe, indicating how concentrated the data is around the linear regression' I am confused here. Which linear regression line are you referring to here? The linear regression between sigma and ECe that is based on the regional dataset and that you use to obtain ECe estimates from EMI measurements, which I would call ECep? Or the linear regression between ECep and the measured ECe from the soil samples? In case of the latter, you are 'correcting' the regional regression equation. At ln 140, you write: 'Lin's CCC measures the agreement between the measured and predicted ECe evaluating how close the linear regression is to the 1:1 relationship' This makes me wonder which regression equation you are talking about here. The regression between ECep and ECe measured? If that is the case, then the RMSE that you mention in ln 139 is not the mean of the squared differences between the measured and predicted ECe but it is the mean of the squared differences between the measured and the ECe's which are predicted using two sequential regressions: first you predict ECe form sigma to obtain ECep and then you make a linear regression between ECep and ECe measured and calculate the RMSE of the deviations between this regression and the measured ECe's. In order to avoid all these confusions, you should write the equations that you used to calculate the RMSE and the R² and you need to define clearly what are the 'predicted' ECe's: the ones obtained using the linear regression based on the regional calibration between sigma and ECe or the predictions by the regression between these predicted ECe's and the measured ECe's?

Please see our detailed answer to the previous question and corresponding revision in this regard.

Ln 154: for clarity, give the depths of 'lower subsoil', 'intermediate subsoil'.

This information was already given in section 2.3 (lines 119 and 120) and also in the legend of Figure 3 (lines 165 and 166).

Ln 165: Figure 3: use the same horizontal axis for the different figures.

Corrected. Thank you.

**Old version (Figure 3)**

[Figure]

**New version (Figure 3)**

[Figure]

Ln 222: 'At location 4 the trend of increasing salinity with depth is accurate in all dates, but it tends to be slightly underestimated. The salinity maps show that salinity increases from non-saline in topsoil to severely-saline in lower subsoil.' How can I see in figure 5 that the trend is underestimated? I do not see that.

We mean that, at location 4, the predicted $EC_e$ (which constitutes the maps) increases with depth, just like the measured $EC_e$ at the sampling sites. However the predicted $EC_e$ (in the background) is lower than the measured $EC_e$, which shows that predicted $EC_e$ is underestimated.

[revised manuscript text omitted]

---

## Editor Decision (ED1)

Thank you for taking up the comments that were made to a previous version of this manuscript. The methods that were used are now clearly presented so that the results can be interpreted better by the readers.

Concerning the $R^2$, you used the definition that quantifies the linear relation between mEC and pEC. This is however different from the more general definition of $R^2$ that quantifies the 1:1 relation between mEC and pEC and that is defined as follows:

$$R^2 = 1 - \frac{\sum_i (mEC_i - pEC_i)^2}{\sum_i (pEC_i - \overline{pEC})^2}$$

You also included statistics for parts of the dataset: $R^2$ per location, soil layer, and measurement time. To some extent, these statistics better unravel the impact of time and location, but, not yet entirely. For location, there is still variation with depth and for layer, there is still variation with location next to the variation over time. I understand that this is done so as to have sufficient data points to calculate statistics. If only the time effect is to be represented, you could look at the relation between the deviations from the mean over time in a certain layer and location and calculate the MSE between the predicted and observed deviations or the $R^2$ between mean and observed deviations. The variance of these deviations would give an impression about the temporal variability of the ECe and how it compares with the spatial variability.

Ln 240: add mS after 128.08

Ln 360: reformulate: At location 4, an increment of salinity along the entire profile is visible during the dry season.

Ln 370: 'This validation resulted in lower prediction ability than that previously resulting from cross-validation' Add maybe the RMSE from the previous cross validation as comparison.

---

## Author Response (AR2)

**"Assessing soil salinity dynamics using time-lapse electromagnetic conductivity imaging" - authors' responses to comments made by Executive Editor Kristof Van Oost**

Dear Executive Editor Kristof Van Oost,

We sincerely appreciate and thank your attention to our manuscript. We revised the manuscript based on the Topical Editor comments. Please find our answers to the Topical Editor comments below. Our answers are placed in blue below each comment. The changes to the manuscript, arising from comments, are written in grey.

Sincerely,

Mohammad Farzamian on behalf of all authors

Authors responses to Topical Editor Jan Vanderborght's comments

Thank you for taking up the comments that were made to a previous version of this manuscript. The methods that were used are now clearly presented so that the results can be interpreted better by the readers.

We are glad that the changes addressed the editor's concerns and contributed to a better version of the paper.

Concerning the R2, you used the definition that quantifies the linear relation between mEC and pEC. This is however different from the more general definition of R2 that quantifies the 1:1 relation between mEC and pEC and that is defined as follows:

$$R^2 = 1 - \frac{\sum_{i=1}^{n}(mEC_{e_i} - pEC_{e_i})^2}{\sum_{i=1}^{n}(pEC_{e_i} - \overline{pEC_e})^2}$$

Agree. We redid the $R^2$ analysis based on the new formula. The text has been reformulated accordingly, and also a new reference was added to the manuscript.

Reference:

Kvålseth, T.O.: Cautionary note about $R^2$. Am. Stat., 39, 279–285, 1985.

**Old version (line 207 - Table 1)**

| Coefficient of determination ($R^2$) | $R^2 = \left( \dfrac{\sum_{i=1}^{n}(mEC_{e_i} - \overline{mEC_e})(pEC_{e_i} - \overline{pEC_e})}{\sqrt{\sum_{i=1}^{n}(mEC_{e_i} - \overline{mEC_e})^2}\sqrt{\sum_{i=1}^{n}(pEC_{e_i} - \overline{pEC_e})^2}} \right)^2$ | Indicates the degree of linearity between predicted and measured data. Ranges from 0 to 1. Above 0.5 is considered satisfactory. |

**New version**

| Coefficient of determination ($R^2$) | $R^2 = 1 - \dfrac{\sum_{i=1}^{n}(mEC_{e_i} - pEC_{e_i})^2}{\sum_{i=1}^{n}(mEC_{e_i} - \overline{mEC_e})^2}$ | Indicates the proportion of the total variation of measured data that is explained by the calibration. Ranges from 0 to 1, although it may be negative values, which indicates an inappropriate calibration (Kvålseth, 1985). Above 0.5 is considered satisfactory. |

**Old version (Table 2)**

|  | $R^2$ |
| --- | --- |
| Global | 0.90 |
| Jan 2018 | 0.93 |
| Jun 2018 | 0.94 |
| Oct 2018 | 0.93 |

**New version**

|  | $R^2$ |
| --- | --- |
| Global | 0.88 |
| Jan 2018 | 0.91 |
| Jun 2018 | 0.90 |
| Oct 2018 | 0.91 |

You also included statistics for parts of the dataset: R2 per location, soil layer, and measurement time. To some extent, these statistics better unravel the impact of time and location, but, not yet entirely. For location, there is still variation with depth and for layer, there is still variation with location next to the variation over time. I understand that this is done so as to have sufficient data points to calculate statistics. If only the time effect is to be represented, you could look at the relation between the deviations from the mean over time in a certain layer and location and calculate the MSE between the predicted and observed deviations or the R2 between mean and observed deviations. The variance of these deviations would give an impression about the temporal variability of the ECe and how it compares with the spatial variability.

We added a new analysis based on the Editor's comment. We calculated RMSE for this analysis, instead of MSE. This is because the Root of MSE (RMSE) has the same units as the quantity plotted and will be easier for readers to interpret it based on the range of variations. This will also make the manuscript more consistent in terms of the statistical analyses presented in Table 1.
The results of this analysis were presented in Table 3 and the manuscript was revised accordingly. The details of changes are given below.

**Old version (line 229)**

[revised manuscript text omitted]

Ln 240: add mS after 128.08

The text was corrected.

**Old version (line 240)**

"128.08 $m^{-1}$"

**New version**

"128.08 mS $m^{-1}$"

Ln 360: reformulate: At location 4, an increment of salinity along the entire profile is visible during the dry season.

The text was reformulated accordingly.

**Old version (line 360)**

"At the rainfed location 4, it is also visible an increment of salinity along the entire profile during the dry season."

**New version**

"At location 4, an increment of salinity along the entire profile is visible during the dry season."

Ln 370: 'This validation resulted in lower prediction ability than that previously resulting from cross- validation' Add maybe the RMSE from the previous cross validation as comparison.

We added the RMSE of the previous validation, calculated using the cross-validation method.

**Old version (line 370)**

[revised manuscript text omitted]